# Class-Grouped Normalized Momentum and Faster Hyperparameter Exploration to Tackle Class Imbalance in Federated Learning

**Haemin Park** [1]  **Diego Klabjan** [1]  **Martin W. Braun** [2]  **Xiuqi Li** [2]  **Balakrishnan Ananthanarayanan** [2]

## Abstract

Class imbalance poses a critical challenge in federated learning (FL), where underrepresented classes suffer from poor predictive performance yet cannot be addressed by standard centralized techniques due to privacy and heterogeneity constraints. We propose FedCGNM (Federated Class-Grouped Normalized Momentum), a client-side optimizer in FL that partitions classes into a small number of groups based on minimum within-group variance, maintains a momentum per group, normalizes each group momentum to unit length, and uses the summation of the normalized group momentums as an update direction. This design both equalizes gradient magnitude across majority and minority groups and mitigates the noise inherent in rare-class gradients. We further provide a theoretical convergence analysis explicitly accounting for time-varying resampling-rates. Additionally, to efficiently optimize these rates in small-client regimes, we introduce FedHOO, an X-armed-bandit (XAB) based algorithm that exploits federated parallelism that evaluates many combinations of two candidate rates per client at linear cost. Empirical evaluation on four public long-tailed benchmarks and a proprietary chip-defect dataset demonstrates that FedCGNM consistently outperforms baselines, with FedHOO yielding further gains in small-scale federations.

## 1. Introduction

Class imbalance remains a major challenge in federated learning (FL) despite prior works (Zhang et al., 2023). When the global label distribution aggregated over all clients is long-tailed, minority classes are underrepresented in training, which degrades their predictive accuracy. Prior works in centralized learning mitigate imbalance through loss reweighting, advanced sampling, or generative augmentation, but these techniques are difficult to deploy in FL because privacy constraints prevent data exchange and synthetic generators are either infeasible or produce unrealistic samples for sparse regimes or domain-specific tasks such as defect detection. For instance, when working with a chip-defect dataset, one of our primary evaluation use case in this paper, synthetic defect images fail to capture the true geometric details of actual defects. To address these challenges, we focus on tackling class imbalance at the optimization level. Unlike data-level augmentations, an optimization-based approach is more generally applicable and remains effective even when data-level tricks are restricted by privacy or the characteristics of datasets.

Another line of work, Per-Class Normalization (PCN), normalizes each class-specific gradient to unit length, preventing any class from dominating an update (Francazi et al., 2023). PCN effectively decreases the loss for all classes and, because it operates directly on gradients, integrates readily with other techniques. However, PCN introduces two critical limitations in multi-class classification: it is heavily affected by *directional noise* (since minority class gradients often misalign with the true descent direction), and the sum of many unit vectors produces a *scaling mismatch* that destabilizes convergence when the number of classes increases. PCN struggles even for a moderate number of classes.

To address these issues, our primary contribution is Federated Class-Grouped Normalized Momentum (FedCGNM). Instead of normalizing all $C$ class-specific gradients, FedCGNM partitions classes into a small number of groups (e.g., majority vs. minority) and applies unit-norm normalization to a momentum vector per group. By reducing the number of normalized vectors from $C$ to just a few, FedCGNM mitigates the scaling-mismatch problem while still giving every group equal magnitude. Furthermore, incorporating momentum attenuates the directional noise from minority classes. An additional benefit concerns client alignment, a primary cause of performance degradation in FL (Dandi et al., 2022). Heterogeneous clients often produce

[1]Department of Industrial Engineering & Management Sciences, Northwestern University, Evanston, IL, USA [2]Intel Corporation, Chandler, AZ, USA. Correspondence to: Haemin Park <haemin.park1@northwestern.edu>.

*Proceedings of the $43^{rd}$ International Conference on Machine Learning*, Seoul, South Korea. PMLR 306, 2026. Copyright 2026 by the author(s).

gradients whose magnitudes per class differ widely, amplifying misalignment. Because FedCGNM forces each group update to have the same unit norm, the aggregated directions across clients become more aligned.

Furthermore, during local training, each client applies resampling to reduce directional noise, so selecting the appropriate sampling-rates is crucial. This choice is nontrivial in FL settings. sampling-rates must be determined jointly across clients, which induces a combinatorial search space, and sampling schedules that fluctuate excessively can destabilize optimization. Our theoretical analysis indicates that stable convergence requires controlling the cumulative variation of sampling-rates across rounds. For these reasons, practical approaches adopt a fixed resampling policy, using a common rate across clients. To find sampling-rates efficiently, we introduce FedHOO as an auxiliary, plug-in algorithm designed specifically for small-scale federations. Based on the X-armed bandit framework, FedHOO exploits the inherent parallelism in FL. In every communication round, FedHOO requests each client to train with only two candidate rates, yet by linearly combining the returned updates the server can infer the validation performance metric for all $2^K$ rate combinations, where $K$ is the number of clients. The method therefore identifies effective sampling-rates early in training, while avoiding an exhaustive sweep of the hyperparameter space.

Across four public benchmarks, our methods consistently outperforms traditional reweighting and sampling baselines, with gains up to 29% over FedAvg combined with resampling. In a large-scale industrial chip defect dataset, our method achieves a 16% improvement over the best baseline. Our main contributions are as follows.

1. **FedCGNM optimizer and variance-aware grouping rule.** We introduce *FedCGNM*, the first client-side optimizer that groups classes, applies unit-norm momentum per group to balance majority and minority influence while reducing noise and scale mismatch. We frame class partitioning as minimization of within-group variance. This rule produces the optimal split found by exhaustive search, yielding a principled yet lightweight grouping strategy.

2. **Convergence analysis.** We incorporate the sampling-rate schedule into the convergence analysis, representing a novel effort combining three components: FL, group normalization, and dynamic sampling-rates. We demonstrate that the sampling-rate schedule impacts convergence.

3. **FedHOO as an auxiliary sampling-rate tuner.** While a grid search for sampling-rates is common, more refined strategies are advantageous when the number of clients is low. The proposed X-armed-bandit-based

exploration scheme is the first optimization based algorithm to determine combinatorial local sampling-rates in FL. The method performs rapid, privacy-preserving search trading off exploration and exploitation.

4. **Strong performance.** We validate our framework across multiple public benchmarks and a real-world industrial semiconductor chip-defect dataset. Across all settings, it consistently shows improvement over strong baselines.

## 2. Related Works

Class imbalance poses a significant challenge in supervised learning, where limited data from minority classes leads to biased models and poor performance on those classes (Chen et al., 2024b; Johnson & Khoshgoftaar, 2019). Common training-level solutions include reweighting, which adjusts learning based on class frequency, optimization-level methods, and resampling, which alters the class distribution in training data.

**Reweighting and Optimization Methods** Optimization-perspective methods generally adjust the learning process by modifying the loss function or the gradient update rule. Most reweighting methods adjust each sample's contribution within the loss function to counteract class imbalance, such as weighted cross-entropy (Aurelio et al., 2019), focal loss (Lin et al., 2017), and class-balanced loss (Cui et al., 2019). Beyond loss-level re-weighting, PCN (Francazi et al., 2023) rescales each class-specific gradient to unit norm, equalizing per-class influence during optimization. He (2024) introduces a technique to adjust the weight of gradient dynamically in a class-incremental learning scenario. They consider reweighting in class-level which works poorly as the number of classes increases, and only consider balance of the gradient magnitude. Different from them, we tackle class imbalance by addressing gradient scaling and directional noise simultaneously.

Several methods have tailored re-weighting to FL setting. CLIMB (Shen et al., 2022) addresses the class imbalance in FL by enforcing the similarity between local empirical losses, while Fed-GraB (Xiao et al., 2023) introduces a self-adjusting gradient balancer that dynamically reweights gradients based on the class distribution inferred from classifier weights. FL Ratio Loss (Wang et al., 2021) builds on centralized Ratio Loss by estimating global class proportions via secure aggregation and adjusting local losses accordingly. FedNoRo (Wu et al., 2023) leverages knowledge distillation and distance-aware aggregation to align client models, and incorporates a logical adjustment mechanism to address both data heterogeneity and class imbalance. Unlike these methods that focus on loss functions or models, we tackle class imbalance at the gradient level, pairing with

a simple resampling technique.

**Resampling and Data Synthesis** Traditional resampling methods balance class balance by removing majority samples (under-sampling) or replicating minority ones (over-sampling) (Carvalho et al., 2025). More recent techniques like SMOTE (Chawla et al., 2002), GAMO (Mullick et al., 2019), and I-GAN (Pan et al., 2024) generate synthetic minority data and show strong empirical performance. However, when minority samples are extremely scarce (Chen et al., 2024a) or synthetic data risks being unrealistic or mislabeled (Alkhawaldeh et al., 2023), such methods become less viable. Thus, this work confines itself to conventional under- and over-sampling without synthetic generation since we focus on those situations.

Choosing how much to re-sample remains an open problem: systematic investigations in centralized deep learning reveal that the optimal under- or over-sampling-rate depends jointly on the dataset size and the severity of class skew (Buda et al., 2018). Curriculum-based schemes, such as Dynamic Curriculum Learning (Wang et al., 2019), further highlight the need to adapt sampling ratios over the course of training rather than fixing them a priori. In the federated setting, Düsing et al. (2024) cast client-side resampling as a tunable policy, optimized to minimize the global loss while respecting privacy constraints. FAST (Wang et al., 2023) advances this idea by viewing each sampling ratio as an arm in a multi-armed-bandit framework, enabling dynamic exploration during training. However, it treats local sampling-rates independently, overlooking the combinatorial nature of FL. Our method adopts this adaptive philosophy, particularly suited to small-scale federations, while addressing the combinatorial optimization challenge.

## 3. Methodology

Consider a federated learning system with $K$ clients for a $C$-class classification task. To cope with class imbalance, we incorporate a resampling strategy, which is a common strategy for tackling class imbalance. Let $\mathcal{D}_k$ denote the original data distribution of client $k$. We define $\mathcal{D}_k(r_k)$ as the data distribution of client $k$ after resampling with a sampling-rate $r_k \geq 0$. If $q_k^{(c)} \neq 0$ is the fraction of samples in class $c$ for client $k$, then under sampling-rate $r_k$ client $k$ resamples class $c$ by a factor $\left(\frac{\max_s q_k^{(s)}}{q_k^{(c)}}\right)^{r_k}$. Consequently, the fraction of class $c$ in a mini-batch changes according to this resampled distribution.

We define the local objective function under resampling as $f_k(x; r_k) = \mathbb{E}_{\xi \sim \mathcal{D}_k(r_k)}[\ell(x; \xi)]$, where $\ell(x; \xi)$ is the sample loss for $\xi$. Given sampling-rates $r = (r_1, \ldots, r_K)$, we define global objective as $f(x; r) = \sum_{k=1}^K p_k f_k(x; r_k)$ where $p_k$ is the weight of client $k$. Furthermore, we assume a

resampling-rate strategy determines the rates $r^{(t)} \in \mathbb{R}^K$ before round $t$ starts. If $\mathcal{F}_t$ is the filtration containing all randomness up to round $t-1$, we assume that $r^{(t)}$ is $\mathcal{F}_t$-measurable.

### 3.1. Federated Class-Grouped-Normalized-Momentum

Per-Class Normalization (PCN) addresses gradient imbalance by normalizing each class-specific gradient and has shown promising performance in imbalanced binary classification. However, it faces a limitation in multi-class settings (see Appendix D.1 for more details). The next-to-be-proposed FedCGNM is designed for multi-class problems and aims to overcome the limitations of PCN.

FedCGNM merges classes into a small number $H \ll C$ of groups (typically majority and minority) and maintains a momentum per group. At communication round $t$, the server broadcasts the global model $x^{(t)}$ and local sampling-rates $r_k^{(t)}$ based on a sampling strategy (see Section 3.4), and each client constructs a partition $\{\mathcal{G}_{k,h}^{(t)}\}_{h=1}^H$ of the classes (see Section 3.2 discerning the grouping rule). If we denote $f_k^{(t)}(x) := f_k(x; r_k^{(t)})$ as the local objective of the client $k$ at round $t$, then it decomposes as $f_k^{(t)} = \sum_{h=1}^H f_{k,h}^{(t)}$, with $f_{k,h}^{(t)}$ being the sum of the loss functions under the resampled distribution over the samples in $\mathcal{G}_{k,h}^{(t)}$.

During local training, client $k$ updates, for each group $h$ and step $i$, the momentum

$$m_{k,h}^{(t,i)} = \beta m_{k,h}^{(t,i-1)} + (1-\beta) g_{k,h}^{(t,i)}, \quad m_{k,h}^{(t,0)} = 0, \quad (1)$$

where $g_{k,h}^{(t,i)}(x) = \nabla f_{k,h}(x; r_k^{(t)}; \xi_{k,h}^{(t,i)})$ is the stochastic gradient computed on samples in mini-batch $\xi_k^{(t,i)}$ drawn solely from group $\mathcal{G}_{k,h}^{(t)}$, and $\beta \in [0,1)$ is the momentum factor. The per-step update direction is obtained by normalizing each momentum with a stabilizing constant $\delta > 0$ and summing across the $H$ groups as

$$x_k^{(t,i)} = x_k^{(t,i-1)} - \eta \sum_{h=1}^H \frac{m_{k,h}^{(t,i)}}{\|m_{k,h}^{(t,i)}\| + \delta} \quad (2)$$

with learning rate $\eta > 0$. The term $\delta$ ensures numerical stability and provides a lower bound on the denominator, which is crucial for theoretical analysis and to avoid numerical instability during training, especially when the gradient norm is small (Yang et al., 2024). After $E$ iterations the client returns $x_k^{(t,E)}$ to the server, which aggregates by $x^{(t+1)} = \sum_k p_k x_k^{(t,E)}$. Operating on a handful of group momenta stabilizes the step norm, suppresses directional noise, and remains computationally efficient even when the number of classes is large.

**Algorithm 1** FedCGNM with Resampling Strategy

**Require:** global rounds $T$, local steps $E$, step size $\eta$, momentum factor $\beta$, client weights $\{p_k\}_{k=1}^K$
1: initialize global model $x^{(0)}$
2: **for** $t = 0, \ldots, T-1$ **do**
3:     Server determines sampling-rates $\{r_k^{(t)}\}_{k=1}^K$ based on resampling strategy (Sec. 3.4).
4:     Server broadcasts $(x^{(t)}, r_k^{(t)})$ to clients.
5:     **for** each client $k$ in parallel **do**
6:         Resample each class $c$ to have $\left(\frac{\max_s q_k^{(s)}}{q_k^{(c)}}\right)^{r_k^{(t)}}$ samples if $q_k^{(c)} \neq 0$. subsequent steps use the resampled data.
7:         Construct groups $\{\mathcal{G}_{k,h}^{(t)}\}_{h=1}^H$ of classes via the grouping rule (Sec. 3.2).
8:         $x_k^{(t,0)} \leftarrow x^{(t)}$;   $m_{k,h}^{(t,0)} \leftarrow 0, \forall h$
9:         **for** $i = 1$ to $E$ **do**
10:           Compute stochastic gradient $g_{k,h}^{(t,i)}$
11:           $m_{k,h}^{(t,i)} \leftarrow \beta m_{k,h}^{(t,i-1)} + (1-\beta)\, g_{k,h}^{(t,i)}, \forall h$
12:           $x_k^{(t,i)} \leftarrow x_k^{(t,i-1)} - \eta \sum_{h=1}^H \frac{m_{k,h}^{(t,i)}}{\|m_{k,h}^{(t,i)}\| + \delta}$
13:         **end for**
14:         Upload $x_k^{(t,E)}$ to server
15:     **end for**
16:     $x^{(t+1)} \leftarrow \sum_k p_k x_k^{(t,E)}$
17: **end for**
18: **return** final model $x^{(T)}$

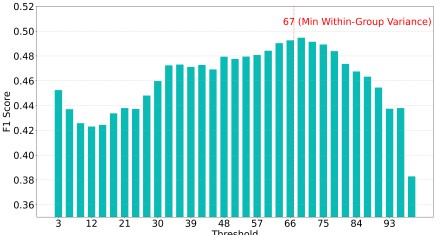

*Figure 1.* Test accuracy on CIFAR-100-LT ($\xi = 20$, $K = 5$) with respect to the number of classes assigned to the minority group. The red line marks the threshold selected by our grouping rule.

### 3.2. Grouping of Classes

We formulate the problem of partitioning $\{1, \ldots, C\}$ into $H$ disjoint groups as one-dimensional variance reduction on the class proportions. For the sake of notation, we omit the dependency on client $k$ and iteration $t$ in the subsequent quantities. Let $q_c$ be the (resampled) proportion of class $c$, with $\sum_{c=1}^C q_c = 1$, and we assume $q_1 \geq \cdots \geq q_C$. We treat $\{q_c\}_{c=1}^C$ as points on the real line and select $H-1$ thresholds to form contiguous groups in which proportions are as similar as possible.

For a partition $G = \{\mathcal{G}_h\}_{h=1}^H$, define the group mass $S_h = \sum_{c \in \mathcal{G}_h} q_c$, group mean mass $\mu_h = S_h/|\mathcal{G}_h|$, and the within-group distribution $w_{c|h} = q_c/S_h$. In a mini-batch of size $B$, let $N_h \sim \mathrm{Binomial}(B, S_h)$ be the sample counts of group $h$ and $N_{h,c}$ be the sample counts of class $c$ in group $h$ based on the distribution. Define the empirical share vector $\hat{\boldsymbol{w}}_h = (\hat{w}_{c|h})_{c \in \mathcal{G}_h}$ with $\hat{w}_{c|h} = N_{h,c}/N_h$, and compare it with the uniform target $u_h = (1/|\mathcal{G}_h|, \ldots, 1/|\mathcal{G}_h|) \in \mathbb{R}^{|\mathcal{G}_h|}$. We define group imbalance vector $\Delta_h = \frac{N_h}{B}(\hat{\boldsymbol{w}}_h - u_h)$.

Taking expectation, we obtain

$$\mathbb{E}\|\Delta_h\|^2 = \left(S_h^2 + \frac{S_h(1-S_h)}{B}\right)\|\hat{\boldsymbol{w}}_h - u_h\|^2 + \frac{S_h}{B}\left(1 - \sum_{c \in \mathcal{G}_h} w_{c|h}^2\right).$$

The optimization problem for grouping is $\min_{\mathcal{G}_h} \mathbb{E}\|\Delta_h\|^2$. We solve this problem heuristically by finding the best threshold that yields groups whose class distributions have minimal within-group variance.

The dominant term of $\mathbb{E}\|\Delta_h\|^2$ is $S_h^2\|\boldsymbol{w}_h - u_h\|^2$ with $O(1/B)$ corrections. We next link this imbalance to the variance of raw proportions. Using $\|\boldsymbol{w}_h - u_h\|^2 = \sum_{c \in \mathcal{G}_h}\left(\frac{q_c}{S_h} - \frac{1}{|\mathcal{G}_h|}\right)^2 = \frac{|\mathcal{G}_h|}{S_h^2}\sigma_h^2$ where $\sigma_h^2 = \frac{1}{|\mathcal{G}_h|}\sum_{c \in \mathcal{G}_h}(q_c - \mu_h)^2$, the dominant term becomes $S_h^2\|\boldsymbol{w}_h - u_h\|^2 = |\mathcal{G}_h|\,\sigma_h^2$. Summing over groups and normalizing per class yields $\frac{1}{C}\sum_{h=1}^H |\mathcal{G}_h|\sigma_h^2 = \sum_{h=1}^H \omega_h \sigma_h^2$ with $\omega_h = |\mathcal{G}_h|/C$, which is the within-group variance objective on the sorted $\{q_c\}_{c=1}^C$. Consequently, selecting $H-1$ thresholds by this strategy produces the partition that minimizes the class-balanced within-group dispersion and, by the argument above, asymptotically minimizes expected per-batch imbalance.

Figure 1 reports the test accuracy obtained when we exhaustively vary the split threshold $\tau_g$, which assigns the $\tau_g$ rarest classes to the minority group and the remaining $C - \tau_g$ classes to the majority group. Accuracy rises to a clear maximum near $\tau_g = 69$ and declines when either too few or too many classes are treated as minority. The red line marks the threshold chosen by our variance-based grouping rule, which, in this example, coincides with the empirical optimum. Additional experiments in Appendix D.2 exhibit the same pattern, confirming that minimizing within-group variance serves as a reliable proxy for an exhaustive threshold search.

In summary, variance-aware grouping via minimizing the within-group variance on empirical class distribution provides a data-driven, lightweight mechanism that, when coupled with normalized momentum, substantially attenuates gradient noise while ensuring balanced directional contributions from majority and minority classes.

## 3.3. Theoretical Convergence Analysis

We analyze the convergence of FedCGNM by explicitly incorporating a resampling-rate schedule $\{r^{(t)}\}$. By modeling the global objective $f^{(t)}(x) = f(x; r^{(t)})$ as a function of these time-varying rates, we can rigorously quantify how the choice of a sampling strategy impacts training dynamics. For the theoretical convergence analysis, we make the following assumptions.

**Assumption 3.1** (Smoothness). *Each local loss function $f_k(\cdot; r)$ is L-smooth for any rate $r$, that is, for all $x, y$ and $r$, we have $\|\nabla f_k(x; r) - \nabla f_k(y; r)\| \leq L \|x - y\|$.*

**Assumption 3.2** (Pathwise boundedness). *There exists $G > 0$ such that for any $t, k$, and $h$, we have $\left\|\nabla f_{k,h}^{(t)}(x^{(t)})\right\| \leq G$.*

**Assumption 3.3** (Unbiasedness and bounded variance). *There exists $\sigma^2 > 0$ such that for any $x, k$, and $r$,*

$$\mathbb{E}\big[\nabla f_k(x; r; \xi_k)\big] = \nabla f_k(x; r), \qquad (3)$$

$$\mathbb{E}\big\|\nabla f_k(x; r; \xi_k) - \nabla f_k(x; r)\big\|^2 \leq \sigma^2. \qquad (4)$$

**Assumption 3.4** (Lipschitz continuity with respect to sampling-rates). *There exists a constant $L_r \geq 0$ such that for any $x$ and any two sampling vectors $r, r'$, we have $|f(x; r) - f(x; r')| \leq L_r \|r - r'\|$.*

**Assumption 3.5.** *Let $\alpha_i := 1 - \beta^i$. There exist $\gamma \in (0, 1]$ and $\kappa > 0$ such that for all $x, k, h, t,$ and $i$, either*

$$\frac{\langle \nabla f_{k,h}^{(t)}(x^{(t)}), m_{k,h}^{(t,i)} \rangle}{\|\nabla f_{k,h}^{(t)}(x^{(t)})\| \|m_{k,h}^{(t,i)}\|} \leq 1 - \gamma \quad or$$

$$\big| \|m_{k,h}^{(t,i)}\| - \|\alpha_i \nabla f_{k,h}^{(t)}(x^{(t)})\| \big| \geq \kappa.$$

Assumption 3.5 asserts that, at each client–class update, the stochastic gradient cannot be both perfectly aligned *and* equal in length to its exponentially averaged momentum. One of the two gaps is virtually guaranteed in practice, and the assumption is strictly weaker than the simultaneous angle-and-norm bounds adopted in earlier analyses of normalized momentum methods. Note that we include the factor $\alpha_i := 1 - \beta^i$ in the norm-gap branch of Assumption 3.5 because $\|m_{k,h}^{(t,i)}\|$ is naturally on the scale of $\|\alpha_i \nabla f_{k,h}(x^{(t)})\|$ rather than $\|\nabla f_{k,h}(x^{(t)})\|$.

The following theorem shows that FedCGNM attains the standard $\mathcal{O}(T^{-1/2})$ stationary-point rate under smoothness and bounded-variance assumptions. A complete proof is given in Appendix A.

**Theorem 3.6.** *Let Assumptions 3.1–3.5 hold and let $\beta = 1 - c\eta$ with a constant $c > 0$ satisfying $c\eta < 1$. Let $\{x^{(t)}\}_{t=1}^{T}$ be the iterates produced by FedCGNM with sampling-rates $\{r^{(t)}\}$, and let each client perform $E \geq 1$ local steps. Let $\Delta_0 = \mathbb{E}[f(x^{(0)}; r^{(0)})] - f_{inf}$, with a finite lower bound $f_{inf}$ of $f(\cdot; \cdot)$. Then we have:*

$$\frac{1}{T} \sum_{t=0}^{T-1} \mathbb{E}\big\|\nabla f(x^{(t)}; r^{(t)})\big\|^2 \leq \frac{\Delta_0}{\eta D_0 E T} + D_1 \eta + D_2 \eta^2$$

$$+ \frac{L_r}{\eta D_0 E T} \sum_{t=0}^{T-1} \sum_{k} p_k \mathbb{E}|r_k^{(t)} - r_k^{(t-1)}|, \tag{5}$$

*where $D_0, D_1$ and $D_2$ are constants defined in the proof. If we define $V_T = \sum_t \sum_k p_k \mathbb{E}|r_k^{(t)} - r_k^{(t-1)}|$, then choosing the step size $\eta = \mathcal{O}(T^{-1/2})$ yields*

$$\min_{t < T} \mathbb{E}[\|\nabla f^{(t)}(x^{(t)})\|^2] \leq \mathcal{O}(T^{-1/2}) + \mathcal{O}(V_T \cdot T^{-1/2}).$$

The second term on the right-hand side of (5) represents the path variation of the sampling-rates. For the algorithm to converge to a stationary point at the standard rate of $\mathcal{O}(T^{-1/2})$, the sampling strategy must ensure that $V_T/\sqrt{T} \to 0$.

## 3.4. Resampling Strategy

The path variation term in Theorem 3.6 implies a fundamental trade-off where aggressive resampling can improve minority class learning, but an indefinite oscillation prevents the global objective from stabilizing. Therefore, a practical resampling strategy must eventually stabilize.

While dynamic sampling-rate search has been explored in centralized curriculum learning (Wang et al., 2019) and in per-client sampling-rate search for FL (Wang et al., 2023), it presents unique challenges in FL due to client heterogeneity and large search-space. If a search algorithm fails to identify sampling rates quickly enough to reduce oscillation during training, convergence may deteriorate. We note, however, that training stability does not strictly guarantee superior test performance. While a full exploration of the optimal balance between these two competing objectives is beyond the scope of this work, our convergence analysis establishes a critical requirement of $V_T$ for practitioners.

For small federations where $K$ is small, we introduce Federated Hierarchical-Optimistic-Optimization (**FedHOO**) as a strategy designed to search effective rates rapidly. This approach leverages the observation that aggregation in FL is linear with respect to client updates, so the server can "mix and match" client updates computed under different local hyperparameter choices. We cast the joint sampling-rate search as an X-armed bandit over a continuous space, and the server follows a HOO-style optimistic rule to balance exploration and exploitation when choosing which rate intervals to probe each round. Here, we outline only the key idea of FedHOO and the full algorithmic details are provided in Appendix B. The following methodology is applicable to

---

**Algorithm 2** FedHOO Resampling Strategy (High-Level)

---

1: **for** each communication round $t$ **do**
2:    **Server:** Sends two probe rates within the search interval to each client $k$.
3:    **Clients:** Compute two local updates in parallel using the probe rates and return them.
4:    **Server:** Synthesizes $2^K$ candidate global models by linearly combining the returned updates.
5:    **Server:** Validates candidates, selects the best model, updates the current model with it, and shrinks search interval using XAB-based exploration rule.
6: **end for**

---

general hyperparameter tuning in FL, but we present it from the perspective of sampling-rates.

To search for joint sampling-rates in FL, the server maintains a plausible interval of sampling-rates for each client $k$. In standard FL, the server can evaluate only one joint hyperparameter configuration per round. In contrast, Fed-HOO enables a much broader exploration by instructing each client to train at two probe rates (e.g., one lower and one upper probe) inside the current interval. Each client performs two local training sessions starting from the same global model using the assigned probe rates and returns the two corresponding client updates, i.e., the model changes produced by local training. Since the server aggregates these updates via a weighted sum, it can construct and validate all $2^K$ models corresponding to every combination of the local rates, select the best performing model for the next starting global model, and shrink each client's interval toward the chosen probe. Iterating this procedure yields rapid rate identification and stabilization.

**Federated validation.** FedHOO does not require centralized validation data. After synthesizing the candidate global models, the server asks clients to evaluate them on their private validation splits. Clients return only scalar validation metrics, such as validation loss or accuracy, and the server aggregates these metrics to select the best candidate model. Thus, FedHOO adds validation overhead but does not require transferring raw validation data to the server. If a server-side hold-out set is available, it can be used as an alternative, but it is not required.

Given that FedHOO requires validating $2^K$ candidates per round, it is designed for small-scale regimes (e.g., $K \leq 5$ or 6). For larger federations where the exponential validation cost becomes prohibitive, we revert to a simpler strategy such as a uniform global rate. Further, in practice, we run FedHOO only in the early rounds to quickly find stable client-specific rates, then fix them for the remaining rounds. Empirically, Appendix E.2 shows that FedHOO reaches stable resampling-rates within a small number of

rounds, whereas a standard exploration strategy that evaluates only one configuration per round fails to stabilize within the same training rounds.

## 4. Experiments

### 4.1. Experiment Settings

Our experiments evaluate FedCGNM and FedHOO on five classification benchmarks: two long-tailed image collections (CIFAR-10-LT and CIFAR-100-LT (Krizhevsky & Hinton, 2009)), two tabular datasets (Adult Income (Becker & Kohavi, 1996) and UNSW-NB15 (Moustafa & Slay, 2015)), and a semiconductor chip-defect dataset. We control the degree of class imbalance via the imbalance rate $\xi$ where larger $\xi$ corresponds to more extreme imbalance. Details of dataset and imbalance settings are provided in Appendix C.

The semiconductor Chip-Defect-Detection (CDD) corpus consists of approximately 780,000 high-resolution (224 × 224 after pre-processing) images captured on select years after 2020 from five factories. The ratio of defect images is 1.7%, with seven defect categories observed across hundreds of product types. Additional heterogeneity distributions are presented in Figure 2 and Figure 4. We split seventy percent of the dataset to training, fifteen percent to validation, and the remaining fifteen percent to test. We report the company's weighted accuracy that balances defect recall and non-defect precision. Some exact counts are withheld in accordance with the non-disclosure agreement. For full details of the implementation, including the selection of hyperparameters, see Appendix C.

We compare FedCGNM against seven alternatives: (1) Fe-dAvg (McMahan et al., 2017) with standard SGD, (2) Fe-dAvg using class-weighted cross-entropy (Aurelio et al., 2019), (3) FedAvg with Ratio Loss (Wang et al., 2021), (4) FedProx (Li et al., 2020), (5) CLIMB (Shen et al., 2022), (6) FedGraB (Xiao et al., 2023), and (7) FedCGN, which adapts Per-Class Normalization to a grouped setting and employs CGN for local optimization. We compare optimization-based imbalance methods because our approach also targets the optimization dynamics, and both our method and these baselines can be readily combined with complementary remedies such as resampling or knowledge distillation. All methods share identical backbones and hyper-parameter schedules, but see Appendix for details of models, initial learning rate, batch size, weight decay and learning-rate scheduler. We use $H = 2$ for all grouping implementations.

Two federation regimes are considered. In the small-scale scenario includes five clients with full participation. In the moderate-scale scenario, twenty clients are available but only half are sampled each round. Public datasets are partitioned IID/Non-IID across clients, where we generate disjoint non-IID client data using a Dirichlet distribution,

*Table 1.* F1 scores on public benchmarks under two federation regimes. "C10" and "C100" abbreviate CIFAR-10 and CIFAR-100, and the trailing numbers denote the imbalance rates. The letter following each baseline denotes the resampling method, with "U" indicating a uniform global sampling-rate. Gray numbers represent the standard deviation across three independent runs.

| Algorithm | C10-20 | C10-100 | C100-20 | C100-100 | Adult-3.17 | Adult-10 | Adult-20 | UNSW-434 |
|---|---|---|---|---|---|---|---|---|
| **K = 5 + IID** | | | | | | | | |
| FedAvg + U | 0.8259 ±0.002 | 0.6177 ±0.003 | 0.4458 ±0.002 | 0.2466 ±0.004 | 0.8431 ±0.001 | 0.9038 ±0.002 | 0.9438 ±0.001 | 0.7705 ±0.001 |
| FedProx + U | 0.8256 ±0.003 | 0.6024 ±0.002 | 0.4483 ±0.002 | 0.2405 ±0.003 | 0.8444 ±0.001 | 0.9052 ±0.002 | 0.9410 ±0.001 | 0.7685 ±0.002 |
| Weighted CE + U | 0.8186 ±0.006 | 0.5379 ±0.007 | 0.3610 ±0.006 | 0.2183 ±0.011 | 0.8394 ±0.002 | 0.9061 ±0.002 | 0.9359 ±0.002 | 0.7223 ±0.006 |
| Ratio Loss + U | 0.8274 ±0.007 | 0.6183 ±0.002 | 0.4402 ±0.004 | 0.2734 ±0.007 | 0.8428 ±0.002 | 0.9065 ±0.002 | 0.9408 ±0.002 | 0.7729 ±0.003 |
| CLIMB + U | 0.8518 ±0.005 | 0.6756 ±0.003 | 0.4439 ±0.003 | 0.2681 ±0.005 | 0.8270 ±0.002 | 0.9052 ±0.002 | 0.9411 ±0.001 | 0.7785 ±0.002 |
| FedGraB + U | 0.8454 ±0.004 | 0.6355 ±0.006 | 0.4321 ±0.004 | 0.2223 ±0.005 | 0.8251 ±0.003 | 0.8878 ±0.004 | 0.9279 ±0.002 | 0.7652 ±0.002 |
| FedCGN + U | 0.8335 ±0.003 | 0.7054 ±0.005 | 0.4925 ±0.004 | 0.3116 ±0.005 | 0.8415 ±0.002 | 0.9099 ±0.001 | 0.9414 ±0.002 | 0.7774 ±0.003 |
| FedCGNM + U | 0.8568 ±0.004 | 0.7432 ±0.002 | 0.4983 ±0.003 | 0.3165 ±0.004 | 0.8455 ±0.002 | 0.9136 ±0.001 | 0.9458 ±0.001 | 0.7804 ±0.002 |
| FedCGNM + FedHOO | **0.8628** ±0.003 | **0.7485** ±0.004 | **0.5021** ±0.004 | **0.3183** ±0.006 | **0.8463** ±0.001 | **0.9151** ±0.002 | **0.9469** ±0.002 | **0.7825** ±0.001 |
| **K = 20 + IID** | | | | | | | | |
| FedAvg + U | 0.7128 ±0.004 | 0.4697 ±0.003 | 0.3544 ±0.002 | 0.1932 ±0.003 | 0.8448 ±0.001 | 0.9125 ±0.001 | 0.9428 ±0.001 | 0.7483 ±0.001 |
| FedProx + U | 0.7015 ±0.003 | 0.4601 ±0.002 | 0.3458 ±0.001 | 0.1884 ±0.002 | 0.8425 ±0.001 | 0.9096 ±0.001 | 0.9416 ±0.002 | 0.7472 ±0.002 |
| Weighted CE + U | 0.6625 ±0.003 | 0.4426 ±0.004 | 0.3213 ±0.001 | 0.1807 ±0.002 | 0.8397 ±0.002 | 0.9081 ±0.002 | 0.9353 ±0.001 | 0.7143 ±0.005 |
| Ratio Loss + U | 0.7159 ±0.002 | 0.4563 ±0.004 | 0.3583 ±0.002 | 0.1950 ±0.003 | **0.8455** ±0.002 | 0.9128 ±0.001 | 0.9441 ±0.002 | 0.7427 ±0.004 |
| CLIMB + U | 0.7663 ±0.003 | 0.4987 ±0.003 | 0.3645 ±0.003 | 0.2091 ±0.003 | 0.8243 ±0.001 | 0.9001 ±0.002 | 0.9292 ±0.002 | 0.7581 ±0.002 |
| FedGraB + U | 0.7846 ±0.003 | 0.5051 ±0.003 | 0.3521 ±0.002 | 0.1657 ±0.004 | 0.8069 ±0.003 | 0.8852 ±0.003 | 0.9314 ±0.002 | 0.7527 ±0.003 |
| FedCGN + U | 0.7915 ±0.003 | 0.5210 ±0.004 | 0.3827 ±0.001 | 0.2138 ±0.003 | 0.8434 ±0.001 | 0.9099 ±0.002 | 0.9458 ±0.001 | 0.7729 ±0.002 |
| FedCGNM + U | **0.8010** ±0.002 | **0.5294** ±0.003 | **0.4051** ±0.002 | **0.2257** ±0.002 | 0.8452 ±0.001 | **0.9145** ±0.002 | **0.9476** ±0.001 | **0.7775** ±0.002 |

*Table 2.* Performance comparison under the Non-IID Scenario for $K = 5$ and $K = 20$.

| Algorithm | C10-20 | C10-100 | C100-20 | C100-100 | Adult-3.17 | Adult-10 | Adult-20 | UNSW-434 |
|---|---|---|---|---|---|---|---|---|
| **K = 5 + Non-IID** | | | | | | | | |
| FedAvg + U | 0.7845 ±0.001 | 0.5372 ±0.003 | 0.4150 ±0.002 | 0.2109 ±0.003 | 0.8369 ±0.001 | 0.8979 ±0.001 | 0.9413 ±0.002 | 0.7513 ±0.001 |
| FedProx + U | 0.7826 ±0.001 | 0.5401 ±0.003 | 0.4089 ±0.004 | 0.2085 ±0.004 | 0.8371 ±0.001 | 0.9001 ±0.001 | 0.9411 ±0.002 | 0.7559 ±0.002 |
| Weighted CE + U | 0.7622 ±0.004 | 0.4961 ±0.006 | 0.3480 ±0.014 | 0.1932 ±0.007 | 0.8334 ±0.003 | 0.8968 ±0.003 | 0.9382 ±0.002 | 0.7372 ±0.002 |
| Ratio Loss + U | 0.7904 ±0.003 | 0.5335 ±0.003 | 0.4093 ±0.004 | 0.1997 ±0.006 | 0.8388 ±0.002 | 0.8965 ±0.003 | 0.9389 ±0.003 | 0.7509 ±0.001 |
| CLIMB + U | 0.8233 ±0.004 | 0.5761 ±0.004 | 0.3852 ±0.006 | 0.2258 ±0.004 | 0.8265 ±0.001 | 0.9017 ±0.002 | 0.9408 ±0.002 | 0.7509 ±0.003 |
| FedGraB + U | 0.8166 ±0.006 | 0.5825 ±0.003 | 0.3733 ±0.008 | 0.2211 ±0.006 | 0.8243 ±0.003 | 0.8914 ±0.002 | 0.9291 ±0.004 | 0.7358 ±0.005 |
| FedCGN + U | 0.8223 ±0.002 | 0.5832 ±0.003 | 0.4210 ±0.002 | 0.2462 ±0.005 | 0.8421 ±0.002 | 0.9054 ±0.001 | 0.9412 ±0.002 | 0.7637 ±0.002 |
| FedCGNM + U | 0.8316 ±0.001 | 0.5896 ±0.004 | 0.4351 ±0.003 | 0.2536 ±0.005 | 0.8460 ±0.002 | 0.9083 ±0.002 | 0.9471 ±0.001 | **0.7778** ±0.001 |
| FedCGNM + FedHOO | **0.8381** ±0.002 | **0.5945** ±0.003 | **0.4427** ±0.005 | **0.2581** ±0.004 | **0.8473** ±0.001 | **0.9094** ±0.002 | **0.9473** ±0.001 | **0.7778** ±0.002 |
| **K = 20 + Non-IID** | | | | | | | | |
| FedAvg + U | 0.6942 ±0.001 | 0.3821 ±0.004 | 0.3347 ±0.002 | 0.1713 ±0.002 | 0.8357 ±0.001 | 0.9023 ±0.002 | 0.9402 ±0.001 | 0.7406 ±0.002 |
| FedProx + U | 0.6958 ±0.002 | 0.3971 ±0.003 | 0.3350 ±0.003 | 0.1686 ±0.002 | 0.8368 ±0.001 | 0.9015 ±0.001 | 0.9389 ±0.001 | 0.7358 ±0.002 |
| Weighted CE + U | 0.5768 ±0.006 | 0.3971 ±0.005 | 0.2903 ±0.007 | 0.1638 ±0.004 | 0.8335 ±0.002 | 0.8980 ±0.001 | 0.9382 ±0.001 | 0.7184 ±0.004 |
| Ratio Loss + U | 0.6596 ±0.004 | 0.3913 ±0.005 | 0.3418 ±0.003 | 0.1641 ±0.004 | 0.8363 ±0.002 | 0.9030 ±0.001 | 0.9421 ±0.001 | 0.7254 ±0.003 |
| CLIMB + U | 0.7263 ±0.003 | 0.4305 ±0.005 | 0.3216 ±0.005 | 0.1828 ±0.003 | 0.8233 ±0.001 | 0.8956 ±0.002 | 0.9315 ±0.002 | 0.7216 ±0.004 |
| FedGraB + U | 0.7391 ±0.003 | 0.4425 ±0.006 | 0.3133 ±0.007 | 0.1634 ±0.005 | 0.8106 ±0.004 | 0.8897 ±0.003 | 0.9309 ±0.002 | 0.7020 ±0.006 |
| FedCGN + U | 0.7361 ±0.002 | 0.4512 ±0.007 | 0.3390 ±0.003 | 0.1785 ±0.003 | 0.8389 ±0.002 | 0.9079 ±0.002 | 0.9425 ±0.001 | 0.7519 ±0.002 |
| FedCGNM + U | **0.7437** ±0.004 | **0.4669** ±0.004 | **0.3463** ±0.003 | **0.1854** ±0.003 | **0.8425** ±0.002 | **0.9095** ±0.002 | **0.9442** ±0.001 | **0.7708** ±0.001 |

Dir($\psi$), with a concentration parameter $\psi$ set to 0.5, as described in (Hsu et al., 2019). The chip corpus is divided according to natural factory boundaries, resulting in highly non-IID client data.

## 4.2. Primary Real-World Evaluation: Chip-Defect Detection (CDD)

The semiconductor CDD task provides the realistic evaluation in our study since this dataset reflects actual production condition, with only 1.7% of samples exhibiting defects. Figure 2 shows that the standard FedAvg baseline stalls at around 73, while other FL baselines provide only marginal improvements. FedCGN pushes performance into 85, and FedCGNM adds another improvement. Using FedHOO as resampling strategy consistently improves training. This demonstrates that variance-aware grouping, per-group momentum, and efficient rate exploration are not only effective on public benchmarks but also critical for real industrial

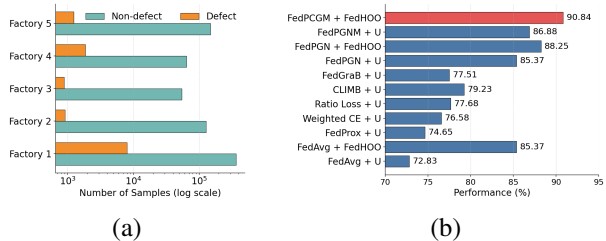

(a)        (b)

*Figure 2.* Analysis of the Chip-Defect Detection (CDD) dataset. (a) The data distribution showing heterogeneity across factories. (b) The comparative test performance of various algorithms on the CDD dataset, with the metric scaled from 0 to 100.

deployments.

## 4.3. Evaluation on Public Datasets

Table 1 summarizes F1 under two federation regimes. Across both federation regimes, FedCGNM outperforms

all baselines by several points on average in all setting. In the small-client setting ($K = 5$), coupling FedCGNM with FedHOO yields the best accuracy in all scenarios. Appendix E.2 further isolates the effect of FedHOO by comparing it against other local exploration strategies . FedHOO consistently improves multiple FL baselines, supporting that its benefit comes from joint rate selection. Even without FedHOO, FedCGNM alone consistently beats competing optimizers. The only exception occurs under mild skew on Adult Income, where Ratio Loss briefly matches FedCGNM, suggesting that simple loss reweighting can suffice when imbalance is mild.

As the number of clients increases from five to twenty, the accuracy decreases for all methods due to the greater heterogeneity. However, FedCGNM exhibits noticeably milder degradation compared to FedCGN, which is the best baseline: on average, FedCGN loses about 14–37% performance, while FedCGNM drops by only 7–32%. This indicates that group momentum not only boosts performance in small federations but also makes optimization more robust to client scaling.

## 4.4. Sensitivity Analysis

**Sensitivity to Imbalance Severity**   On the multi-class image benchmarks (CIFAR-10 and CIFAR-100), all methods show declining F1 scores as the class distribution becomes more imbalanced, but FedCGNM's drop is noticeably milder. By contrast, on the Adult Income task, FedCGNM offers only a slight improvement, reflecting the relative ease of a binary prediction problem.

*Table 3.* F1 score for FedCGNM with different group counts $H$.

| Dataset | $K$ | $H = 2$ | $H = 3$ | $H = 4$ |
|---|---|---|---|---|
| CF10-LT20 | 5 | 0.8565 | 0.8400 | 0.8341 |
| | 20 | 0.8010 | 0.7916 | 0.7148 |
| CF10-LT100 | 5 | 0.7432 | 0.7328 | 0.7171 |
| | 20 | 0.5294 | 0.5185 | 0.4957 |
| CF100-LT20 | 5 | 0.4983 | 0.4941 | 0.4485 |
| | 20 | 0.4051 | 0.3916 | 0.3547 |
| CF100-LT100 | 5 | 0.3165 | 0.3121 | 0.2721 |
| | 20 | 0.2257 | 0.2253 | 0.2007 |

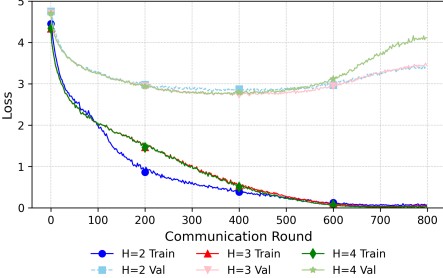

*Figure 3.* Training and validation loss on CIFAR-100-LT ($\xi = 20$, $K = 20$) for FedCGNM with different group counts $H$.

*Table 4.* Accuracy (%) across different $\beta$ values.

| Dataset | K | 0 | 0.1 | 0.3 | 0.5 | 0.7 | 0.9 |
|---|---|---|---|---|---|---|---|
| C10-20 | 5 | 83.35 | 83.72 | 84.21 | **85.65** | 84.27 | 83.93 |
| | 20 | 79.15 | 79.55 | 79.55 | **80.10** | 79.50 | 79.54 |
| C10-100 | 5 | 70.54 | 71.11 | 71.11 | **74.32** | 71.11 | 71.11 |
| | 20 | 52.10 | 52.34 | 52.35 | **52.94** | 52.65 | 52.68 |
| C100-20 | 5 | 49.25 | 49.33 | 49.35 | **49.83** | 49.46 | 49.76 |
| | 20 | 38.27 | 39.16 | 39.52 | **40.51** | 39.84 | 40.24 |
| C100-100 | 5 | 31.16 | 31.22 | 31.08 | **31.65** | 30.95 | 30.03 |
| | 20 | 21.38 | 21.92 | 22.22 | **22.57** | 22.56 | 22.48 |

**Effect of Grouping and Number of Groups**   We examine the impact of the number of groups using training and validation losses for $H = 2, 3, 4$ in Figure 3 and Table 3. For additional results, see Appendix D.3). With two groups, validation loss remains lowest and most stable, whereas three or four groups lead to earlier increases in validation loss despite continued training loss reduction, indicating accelerated overfitting. On CIFAR-100, $H = 3$ occasionally yields slightly better accuracy, but the gap is marginal, so two groups offer the best trade-off between balanced class influence and generalization. Beyond the number of groups, our grouping strategy based on minimizing within-group variance provides clear benefits compared to naive half-splits: on CIFAR-10-LT20 with $K = 5$, our rule achieves an F1 score of 0.8565 versus 0.8382, and on CIFAR-100-LT20 with $K = 5$, it improves from 0.4339 to 0.4983. These gains confirm that variance-aware grouping better balances class contributions and yields superior generalization across datasets.

**Sensitivity to Group Momentum Factor**   We perform an ablation study on the group momentum factor $\beta$ across multiple datasets and client settings. The results in Table 4 show that very small ($\beta = 0$, which is FedCGN) or very large ($\beta = 0.9$) values generally underperform, while moderate values yield the strongest results. For example, on CIFAR-10-LT20 with $K = 5$, accuracy improves from 83.35 at $\beta = 0$ to a peak of 85.68 at $\beta = 0.5$, and on CIFAR-100-LT20 with $K = 20$, performance rises from 38.27 to 40.51 at the same setting. Similar trends appear consistently across datasets, indicating that moderate momentum factors strike the right balance between stability and adaptivity, thereby offering the best overall performance.

Additional experiments (Appendix E) confirm that FedHOO accelerates sampling-rate search compared to standard HOO, and FedCGNM maintains its advantage under large-client federations. These results demonstrate the robustness and generality of our approach across diverse and challenging federated learning conditions.

# 5. Conclusion

We introduced FedCGNM, a client-side optimizer that balances majority and minority class influence by grouping labels and applying group-wise normalized momentum, and FedHOO, a federated exploration strategy that exploits FL parallelism to efficiently tune client sampling rates without exhaustive search (in small-$K$ regimes). We develop a convergence analysis that accounts for dynamic sampling-rate schedules. Experiments on multiple public benchmarks and a large-scale industrial chip-defect dataset demonstrate consistent improvements over competing baselines. Together, these components form a practical framework for mitigating global class imbalance in privacy-preserving federated settings.

# Acknowledgements

The authors express sincere gratitude to Intel Corporation for their generous financial support, which was instrumental in completion of this research. Their commitment to advancing innovation has significantly contributed to the success of this project.

# Impact Statement

This paper presents work whose goal is to advance the field of Machine Learning. There are many potential societal consequences of our work, none which we feel must be specifically highlighted here.

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

# A. Proof

## A.1. Supplementary Lemmas

**Lemma A.1.** *Let Assumptions 3.2 and 3.5 hold. For every client $k$, group index $h$, global round $t$, and local step $i$,*

$$\left\| \alpha_i \nabla f_{k,h}^{(t)}(x^{(t)}) - \frac{m_{k,h}^{(t,i)}}{\|m_{k,h}^{(t,i)}\| + \delta} \right\|^2 \leq \rho \left\| \alpha_i \nabla f_{k,h}^{(t)}(x^{(t)}) - m_{k,h}^{(t,i)} \right\|^2, \qquad \rho := \max\left\{ \frac{2}{\gamma} \max\{1, \delta^{-2}\}, \frac{(G+1)^2}{\kappa^2} \right\}. \quad (6)$$

*Proof.* Fix $(k, h, t, i)$ and set $a := \alpha_i \nabla f_{k,h}^{(t)}(x^{(t)})$, $b := m_{k,h}^{(t,i)}$, $u := b/(\|b\| + \delta)$, $d := \|a\|$, $r := \|b\|$, and $\theta := \angle(a, b)$. Note that $d = \|\alpha_i \nabla f_{k,h}^{(t)}(x^{(t)})\| \leq \alpha_i G \leq G$ by Assumption 3.2. Also define $s := \|u\| = r/(r + \delta) \leq 1$.

Define

$$R(d, r, \theta) := \frac{\|a - u\|^2}{\|a - b\|^2} = \frac{d^2 + s^2 - 2ds \cos\theta}{d^2 + r^2 - 2dr \cos\theta}.$$

We will find an upper bound of $R(d, r, \theta)$ in the two mutually exclusive cases of Assumption 3.5.

(i) *Angle gap:* $1 - \cos\theta \geq \gamma$ implies

$$\|a - b\|^2 = d^2 + r^2 - 2dr \cos\theta \geq d^2 + r^2 - 2dr(1 - \gamma) = (d - r)^2 + 2\gamma dr = \gamma(d^2 + r^2) + (1 - \gamma)(d - r)^2 \geq \gamma(d^2 + r^2).$$

For the numerator, by $\|x - y\|^2 \leq 2\|x\|^2 + 2\|y\|^2$ and $s = r/(r + \delta) \leq r/\delta$,

$$\|a - u\|^2 \leq 2d^2 + 2s^2 \leq 2d^2 + \frac{2r^2}{\delta^2} \leq 2\max\{1, \delta^{-2}\}(d^2 + r^2).$$

Therefore

$$R(d, r, \theta) \leq \frac{2}{\gamma} \max\{1, \delta^{-2}\}.$$

(ii) *Norm gap:* $|r - d| \geq \kappa$ yields

$$\|a - b\|^2 \geq (r - d)^2 \geq \kappa^2.$$

Also $\|a - u\| \leq \|a\| + \|u\| \leq d + 1 \leq G + 1$, so

$$R(d, r, \theta) \leq \frac{(G + 1)^2}{\kappa^2}.$$

Hence $R(d, r, \theta) \leq \rho$ with $\rho$ given in (6); substituting $R$ completes the proof. $\qquad \square$

**Corollary A.2.** *Under Assumption 3.2 and 3.5, for every client $k$, class $h$, round $t$, and local step $i$,*

$$\mathbb{E}\left\| \alpha_i \nabla f_{k,h}^{(t)}(x^{(t)}) - \frac{m_{k,h}^{(t,i)}}{\|m_{k,h}^{(t,i)}\| + \delta} \right\|^2 \leq \rho \, \mathbb{E}\left\| \alpha_i \nabla f_{k,h}^{(t)}(x^{(t)}) - m_{k,h}^{(t,i)} \right\|^2, \qquad \forall k, h, t, i. \quad (7)$$

*Proof.* Because inequality (6) is valid pointwise, it holds for every outcome of the algorithm's randomness. Taking expectations on both sides preserves the inequality, yielding (7). $\qquad \square$

**Lemma A.3.** *Let $\{x_k^{(t,s)}\}_{s=0}^{E}$ and $m_{k,h}^{(t,i)}$ be the local iterate and momentum of client $k$ in communication round $t$, produced by FedCGNM. Under Assumption 3.1-3.3, for any group $h \in \{1, \ldots, H\}$, and local step $i \in \{1, \ldots, E\}$,*

$$\mathbb{E}\left\| \alpha_i \nabla f_{k,h}^{(t)}(x_k^{(t,0)}) - m_{k,h}^{(t,i)} \right\|^2 \leq 2i^2 L^2 H^2 \eta^2 + 4i(1 - \beta)^2 \sigma^2. \quad (8)$$

*Proof.* Suppress fixed indices $(k, h, t)$ and define

$$g_1 := \nabla f_{k,h}^{(t)}(x_k^{(t,0)}), \qquad g_s := \nabla f_{k,h}^{(t)}(x_k^{(t,s-1)}), \qquad m_s := m_{k,h}^{(t,s)}, \qquad g_s^{\text{st}} := g_{k,h}^{(t,s)}(x_k^{(t,s-1)}).$$

Unrolling the momentum recursion yields

$$m_i = (1 - \beta) \sum_{j=1}^{i} \beta^{i-j} g_j^{\text{st}}.$$

With $w_j := (1 - \beta)\beta^{i-j}$, we have $\sum_{j=1}^{i} w_j = 1 - \beta^i = \alpha_i$, hence

$$\alpha_i g_1 - m_i = \sum_{j=1}^{i} w_j (g_1 - g_j^{\text{st}}) = \underbrace{\sum_{j=1}^{i} w_j (g_1 - g_j)}_{=:A} + \underbrace{\sum_{j=1}^{i} w_j (g_j - g_j^{\text{st}})}_{=:B}.$$

Therefore, $\|A + B\|^2 \leq 2\|A\|^2 + 2\|B\|^2$ gives

$$\mathbb{E}\|\alpha_i g_1 - m_i\|^2 \leq 2\mathbb{E}\|A\|^2 + 2\mathbb{E}\|B\|^2. \tag{9}$$

**(1) Drift term $A$.** By Jensen inequality ($w_j \geq 0$ and $\sum_{j=1}^{i} w_j \leq 1$), we have

$$\mathbb{E}\|A\|^2 = \mathbb{E}\left\|\sum_{j=1}^{i} w_j (g_1 - g_j)\right\|^2 \leq \sum_{j=1}^{i} w_j \mathbb{E}\|g_1 - g_j\|^2.$$

Since $\|x^{(s)} - x^{(s-1)}\| \leq \eta H$, we have $\|x^{(j-1)} - x^{(0)}\| \leq (j - 1)\eta H$, and by $L$-smoothness,

$$\|g_1 - g_j\| \leq L\|x^{(0)} - x^{(j-1)}\| \leq L(j - 1)\eta H, \quad \Rightarrow \quad \|g_1 - g_j\|^2 \leq L^2 H^2 (j - 1)^2 \eta^2.$$

Thus

$$\mathbb{E}\|A\|^2 \leq \sum_{j=1}^{i} w_j L^2 H^2 (j - 1)^2 \eta^2 \leq i^2 L^2 H^2 \eta^2 \sum_{j=1}^{i} w_j \leq i^2 L^2 H^2 \eta^2.$$

**(2) Noise term $B$.** Let $\varepsilon_j := g_j^{\text{st}} - g_j$. Then $\mathbb{E}[\varepsilon_j \mid \mathcal{F}_{j-1}] = 0$ and $\mathbb{E}\|\varepsilon_j\|^2 \leq \sigma^2$. We have $B = -\sum_{j=1}^{i} w_j \varepsilon_j$ and, for $j < \ell$,

$$\mathbb{E}\langle \varepsilon_j, \varepsilon_\ell \rangle = \mathbb{E}\Big[\mathbb{E}[\langle \varepsilon_j, \varepsilon_\ell \rangle \mid \mathcal{F}_{\ell-1}]\Big] = \mathbb{E}\big[\langle \varepsilon_j, \mathbb{E}[\varepsilon_\ell \mid \mathcal{F}_{\ell-1}]\rangle\big] = 0,$$

since $\varepsilon_j$ is $\mathcal{F}_{\ell-1}$-measurable. Hence cross terms vanish and

$$\mathbb{E}\|B\|^2 = \sum_{j=1}^{i} w_j^2 \mathbb{E}\|\varepsilon_j\|^2 \leq \sigma^2 \sum_{j=1}^{i} w_j^2.$$

Now

$$\sum_{j=1}^{i} w_j^2 = (1 - \beta)^2 \sum_{r=0}^{i-1} \beta^{2r} = (1 - \beta)^2 \frac{1 - \beta^{2i}}{1 - \beta^2} = \frac{1 - \beta}{1 + \beta}(1 - \beta^{2i}) \leq 2i(1 - \beta)^2.$$

Therefore $\mathbb{E}\|B\|^2 \leq 2i(1 - \beta)^2 \sigma^2$.

**Combine.** Substitute the two bounds into (9) to obtain

$$\mathbb{E}\|\alpha_i g_1 - m_i\|^2 \leq 2i^2 L^2 H^2 \eta^2 + 4i(1 - \beta)^2 \sigma^2,$$

which is (8). $\qquad\square$

**Lemma A.4.** *Let $f^{(t)}(x) = f(x; r^{(t)}) = \sum_{k=1}^{K} p_k f_k(x; r_k^{(t)})$ with $p_k \geq 0$ and $\sum_k p_k = 1$. Then for any iterates $x^{(t)}$ and $x^{(t+1)}$,*

$$f^{(t+1)}(x^{(t+1)}) - f^{(t)}(x^{(t)}) \leq \left( f^{(t)}(x^{(t+1)}) - f^{(t)}(x^{(t)}) \right) + L_r \sum_{k=1}^{K} p_k \left| r_k^{(t+1)} - r_k^{(t)} \right|. \tag{10}$$

*Proof.* First, for any fixed $x$,

$$\left| f^{(t+1)}(x) - f^{(t)}(x) \right| = \left| \sum_{k=1}^{K} p_k \left( f_k^{(t)}(x; r_k^{(t+1)}) - f_k^{(t)}(x; r_k^{(t)}) \right) \right|$$

$$\leq \sum_{k=1}^{K} p_k \left| f_k^{(t)}(x; r_k^{(t+1)}) - f_k^{(t)}(x; r_k^{(t)}) \right| \leq L_r \sum_{k=1}^{K} p_k |r_k^{(t+1)} - r_k^{(t)}|,$$

where we used $p_k \geq 0$ and Assumption 3.4. In particular, applying this bound at $x = x^{(t+1)}$ gives

$$f^{(t+1)}(x^{(t+1)}) \leq f^{(t)}(x^{(t+1)}) + L_r \sum_{k=1}^{K} p_k |r_k^{(t+1)} - r_k^{(t)}|.$$

Subtracting $f^{(t)}(x^{(t)})$ from both sides yields (10). □

### A.2. Proof of Theorem 3.6

*Proof.* For each client $k$ and communication round $t$, the local update can be expressed as

$$x_k^{(t,E)} - x_k^{(t,0)} = -\eta \sum_{i=1}^{E} \sum_{h=1}^{H} \frac{m_{k,h}^{(t,i)}}{\|m_{k,h}^{(t,i)}\|} = -\eta E \Delta_k^{(t)}, \tag{11}$$

where we define $\Delta_k^{(t)} = \frac{1}{E} \sum_{i=1}^{E} \sum_{h=1}^{H} \frac{m_{k,h}^{(t,i)}}{\|m_{k,h}^{(t,i)}\|}$ as the average local update of client $k$ in the communication round $t$.

By $L$-smoothness assumption,

$$\mathbb{E}\left[ f^{(t)}(x^{(t+1)}) \right] \leq \mathbb{E}\left[ f^{(t)}(x^{(t)}) \right] + \mathbb{E}\left[ \langle \nabla f^{(t)}(x^{(t)}), x^{(t+1)} - x^{(t)} \rangle \right] + \frac{L}{2} \mathbb{E} \|x^{(t+1)} - x^{(t)}\|^2. \tag{12}$$

For the third term, by using Jensen's Inequality, we have

$$\mathbb{E}\|x^{(t+1)} - x^{(t)}\|^2 = \mathbb{E}\left\| \sum_k p_k \cdot \eta \sum_{i=1}^{E} \sum_{h=1}^{H} \frac{m_{k,h}^{(t,i)}}{\|m_{k,h}^{(t,i)}\|} \right\|^2$$

$$\leq \eta^2 \sum_k p_k \mathbb{E}\left\| \sum_{i=1}^{E} \sum_{h=1}^{H} \frac{m_{k,h}^{(t,i)}}{\|m_{k,h}^{(t,i)}\|} \right\|^2 \tag{13}$$

$$\leq \eta^2 \sum_k p_k E^2 H^2$$

$$\leq \eta^2 E^2 H^2.$$

By plugging (11) and (13) into (12), we have

$$\mathbb{E}\left[ f^{(t)}(x^{(t+1)}) \right] - \mathbb{E}\left[ f^{(t)}(x^{(t)}) \right] \leq -\eta E \, \mathbb{E}\left[ \langle \nabla f^{(t)}(x^{(t)}), \sum_k p_k \Delta_k^{(t)} \rangle \right] + \frac{\eta^2 E^2 H^2 L}{2}. \tag{14}$$

We define $A = \sum_{i=1}^{E} \frac{\alpha_i}{E} = 1 - \frac{\beta(1-\beta^E)}{E(1-\beta)}$ as the sum of $\alpha_i/E$. For some $\alpha > 0$, we apply Young's Inequality $\langle a, b \rangle \leq \frac{\alpha}{2}\|a\|^2 + \frac{1}{2\alpha}\|b\|^2$, then we have

$$
\begin{aligned}
\mathbb{E}\left[f^{(t)}(x^{(t+1)})\right] - \mathbb{E}\left[f^{(t)}(x^{(t)})\right] &\leq -\eta E \, \mathbb{E}\left[\langle \nabla f^{(t)}(x^{(t)}), \sum_k p_k \Delta_k^{(t)} \rangle\right] + \frac{\eta^2 E^2 H^2 L}{2} \\
&\leq -\eta E \, \mathbb{E}\left[\langle \nabla f^{(t)}(x^{(t)}), \sum_k p_k \Delta_k^{(t)} - A\nabla f^{(t)}(x^{(t)}) + A\nabla f^{(t)}(x^{(t)}) \rangle\right] + \frac{\eta^2 E^2 H^2 L}{2} \\
&\leq -\eta A E \, \mathbb{E}\|\nabla f^{(t)}(x^{(t)})\|^2 - \eta E \, \mathbb{E}\left[\langle \nabla f^{(t)}(x^{(t)}), \sum_k p_k \Delta_k^{(t)} - A\nabla f^{(t)}(x^{(t)}) \rangle\right] + \frac{\eta^2 E^2 H^2 L}{2} \\
&\leq -\eta A E \, \mathbb{E}\|\nabla f^{(t)}(x^{(t)})\|^2 + \frac{\alpha \eta E}{2}\mathbb{E}\|\nabla f^{(t)}(x^{(t)})\|^2 \\
&\quad + \frac{\eta E}{2\alpha}\mathbb{E}\left\|A\nabla f^{(t)}(x^{(t)}) - \sum_k p_k \Delta_k^{(t)}\right\| + \frac{\eta^2 E^2 H^2 L}{2} \\
&= -\eta E\left(A - \frac{\alpha}{2}\right)\mathbb{E}\|\nabla f^{(t)}(x^{(t)})\|^2 + \frac{\eta E}{2\alpha}\mathbb{E}\left\|A\nabla f^{(t)}(x^{(t)}) - \sum_k p_k \Delta_k^{(t)}\right\| + \frac{\eta^2 E^2 H^2 L}{2}.
\end{aligned}
\tag{15}
$$

Rewrite the discrepancy in (15) as

$$
\begin{aligned}
A\,\nabla f^{(t)}(x^{(t)}) - \sum_k p_k \Delta_k^{(t)} &= \sum_{k=1}^{K} p_k \left(A\nabla f_k^{(t)}(x^{(t)}) - \Delta_k^{(t)}\right) \\
&= \sum_{k=1}^{K} p_k \left(\frac{1}{E}\sum_{i=1}^{E}\alpha_i \nabla f_k^{(t)}(x^{(t)}) - \frac{1}{E}\sum_{i=1}^{E}\sum_{h=1}^{H}\frac{m_{k,h}^{(t,i)}}{\|m_{k,h}^{(t,i)}+\delta\|}\right) \\
&= \frac{1}{E}\sum_{k=1}^{K} p_k \sum_{i=1}^{E}\sum_{h=1}^{H}\left(\alpha_i \nabla f_{k,h}^{(t)}(x^{(t)}) - \frac{m_{k,h}^{(t,i)}}{\|m_{k,h}^{(t,i)}+\delta\|}\right)
\end{aligned}
\tag{16}
$$

Applying Cauchy–Schwarz and then Corollary A.2,

$$
\begin{aligned}
\mathbb{E}\left\|A\,\nabla f^{(t)}(x^{(t)}) - \sum_k p_k \Delta_k^{(t)}\right\|^2 &\leq \frac{H}{E}\sum_k p_k \sum_i \sum_h \mathbb{E}\left\|\alpha_i \nabla f_{k,h}^{(t)}(x^{(t)}) - \frac{m_{k,h}^{(t,i)}}{\|m_{k,h}^{(t,i)}+\delta\|}\right\|^2 \\
&\leq \frac{\rho H}{E}\sum_k p_k \sum_i \sum_h \mathbb{E}\left\|\alpha_i \nabla f_{k,h}^{(t)}(x^{(t)}) - m_{k,h}^{(t,i)}\right\|^2.
\end{aligned}
\tag{17}
$$

Combining the descent bound (15), discrepency bound (17) yields, for every global round $t$,

$$
\begin{aligned}
\mathbb{E}\left[f^{(t)}(x^{(t+1)})\right] - \mathbb{E}\left[f^{(t)}(x^{(t)})\right] &\leq -\eta E\left(A - \frac{\alpha}{2}\right)\mathbb{E}\|\nabla f^{(t)}(x^{(t)})\|^2 + \frac{\eta^2 E^2 H^2 L}{2} \\
&\quad + \frac{\eta E}{2\alpha}\left(\frac{\rho H}{E}\sum_k p_k \sum_{i=1}^{E}\sum_{h=1}^{H}\mathbb{E}[\|\alpha_i \nabla f_{k,h}^{(t)}(x^{(t)}) - m_{k,h}^{(t,i)}\|^2]\right).
\end{aligned}
\tag{18}
$$

We apply the moment-difference bound of Lemma A.3, then we have

$$\mathbb{E}\Big[f^{(t)}(x^{(t+1)})\Big] - \mathbb{E}\Big[f^{(t)}(x^{(t)})\Big] \leq -\eta E(A - \frac{\alpha}{2})\mathbb{E}\|\nabla f^{(t)}(x^{(t)})\| + \frac{2\rho(1-\beta)^2\sigma^2 E^2 H^2}{\alpha}\eta + \frac{E^2 H^2 L}{2}\eta^2 + \frac{\rho E^3 H^4 L^2}{\alpha}\eta^3. \tag{19}$$

By taking expectations from Lemma A.4, we have

$$\mathbb{E}[f^{(t+1)}(x^{(t+1)})] - \mathbb{E}[f^{(t)}(x^{(t)})] \leq \mathbb{E}[f^{(t)}(x^{(t+1)})] - \mathbb{E}[f^{(t)}(x^{(t)})] + L_r \sum_k p_k \mathbb{E}|r_k^{(t+1)} - r_k^{(t)}|. \tag{20}$$

We define $V_T := \sum_{t=0}^{T-1} \sum_k p_k \mathbb{E}|r_k^{(t+1)} - r_k^{(t)}|$. Set $1 - \beta = c\eta$ for some $c > 0$ satisfying $c\eta < 1$. Applying (20), re-arrainging (19) and summation over $T$ yields

$$\frac{1}{T}\sum_{t=0}^{T-1} \mathbb{E}\big\|\nabla f^{(t)}(x^{(t)})\big\|^2 \leq \frac{\mathbb{E}[f^{(0)}(x^{(0)})] - \mathbb{E}[f^{(T)}(x^{(T)})]}{\eta D_0 E T} + D_1\eta + D_2\eta^2 + \frac{L_r}{\eta D_0 E T} \cdot V_T, \tag{21}$$

where the deterministic coefficients are

$$D_0 := A - \frac{\alpha}{2},$$
$$D_1 := \frac{EH^2 L}{2\left(A - \frac{\alpha}{2}\right)},$$
$$D_2 := \frac{\rho EH^2}{\alpha\left(A - \frac{\alpha}{2}\right)}(2c^2\sigma^2 + EH^2 L^2).$$

By the assumption that the objective function is bounded from below, we have $\mathbb{E}[f(x^{(T)})] \geq f_{\inf}$. Consequently, $\mathbb{E}[f^{(0)}(x^{(0)})] - \mathbb{E}[f^{(T)}(x^{(T)})] \leq \mathbb{E}[f^{(0)}(x^{(0)})] - f_{\inf}$. We choose $\alpha = 2A - C > 0$ for some constant C. Substituting $\eta = \eta_0 T^{-1/2}$ into (21), we analyze each term on the right-hand side.

$$\frac{\mathbb{E}[f^{(0)}(x^{(0)})] - \mathbb{E}[f^{(T)}(x^{(T)})]}{\eta D_0 E T} \leq \frac{\mathbb{E}[f^{(0)}(x^{(0)})] - f_{\inf}}{\eta C E T} = \mathcal{O}(T^{-1/2}),$$

$$D_1\eta = \frac{EH^2 L}{2(A - \frac{\alpha}{2})} \cdot \eta = \frac{EH^2 L}{2C} \cdot \eta = \mathcal{O}(T^{-1/2}),$$

$$D_2\eta^2 = \frac{\rho EH^2}{\alpha(A - \frac{\alpha}{2})}(2c^2\sigma^2 + EH^2 L^2) \cdot \eta^2$$

$$= \frac{\rho EH^2}{(2A - C)C}(2c^2\sigma^2 + EH^2 L^2) \cdot \eta^2 = \mathcal{O}(T^{-1}),$$

$$\frac{L_r}{\eta D_0 E T} \cdot V_T = \frac{L_r}{\eta C E T} \cdot V_T = \mathcal{O}(V_T \cdot T^{-1/2})$$

Therefore, the RHS of (21) is $\mathcal{O}(T^{-1/2}) + \mathcal{O}(V_T \cdot T^{-1/2})$. Taking the minimum over $t = 0, \ldots, T - 1$ on the left and observing that each term is nonnegative gives the same upper bound for $\min_{t<T} \mathbb{E}[\|\nabla f^{(t)}(x^{(t)})\|^2]$. Hence,

$$\min_{t<T} \mathbb{E}[\|\nabla f^{(t)}(x^{(t)})\|^2] \leq \mathcal{O}(T^{-1/2}) + \mathcal{O}(V_T \cdot T^{-1/2}), \tag{22}$$

which completes the proof of Theorem 3.6. $\qquad\square$

$\square$

# B. Details of FedHOO

This section formalizes FedHOO as a continuous hyperparameter search problem over client-wise resampling-rates. We first cast rate selection as an X-armed bandit on the domain $\mathcal{X} = [r_{\min}, r_{\max}]^K$, and briefly review how classical hierarchical optimistic optimization (HOO) navigates this space via a tree-based partition. We then introduce FedHOO, which retains HOO's hierarchical structure but exploits the linear aggregation in federated learning to evaluate $2^K$ joint rate configurations per round from only two local runs per client, enabling substantially faster exploration and stabilization in small-client regimes.

## B.1. FedHOO Algorithm

A possible way to find sampling-rates is to view the problem through the lens of the X-armed bandit (XAB) over the continuous domain $\mathcal{X} = [r_{\min}, r_{\max}]^K$. An arm $r = (r_1, \ldots, r_K) \in \mathcal{X}$ specifies client rates, and pulling $r$ consists of executing one FL round where client $k$ trains with $r_k$, aggregating the local models, and evaluating the aggregated model. The reward $f(r)$ is the performance of the resulting aggregated model, and the learner must balance exploration and exploitation to identify a near-optimal $r \in \mathcal{X}$. Hierarchical Optimistic Optimization (HOO, (Bubeck et al., 2011)) addresses XAB by creating a hierarchical, tree-based partition of the search space $\mathcal{X}$. The algorithm iteratively navigates this tree, selecting the most promising subregions to explore based on an optimistic estimate of their potential reward.

However, standard XAB methods (e.g., exhaustive search or HOO) converge too slowly under FL's limited rounds and early sensitivity to hyperparameters. We introduce FedHOO, which exploits the parallelism of FL and is suitable for a small number of clients. FedHOO retains HOO's tree of boxes and enables much faster exploration by letting clients perform local training with two rates and exploring all combinations of local rates for validations.

The space $\mathcal{X} = [r_{\min}, r_{\max}]^K$ is organized as a $2^K$-ary tree. The root covers the entire search space, and each child $\nu$ halves its parent's range, and the midpoint $c(\nu)$ of node $\nu$'s interval acts as its representative point. Each node $\nu$ in depth $h$ stores a box $I(\nu) = \{[L_k(\nu), U_k(\nu)]\}_{k=1}^K$, an evaluation count $N(\nu)$, a running reward estimate $V(\nu)$, and an optimistic score $B(\nu)$.

At round $t$, the server selects the leaf $\nu$ with largest $B(\nu)$. Let $\mathcal{I}(\nu) = \prod_k [L_k, U_k]$ be its box and $h$ be its depth in the tree. For each client $k$, the server sends two rates $r_k^L = (3L_k + U_k)/4$, $r_k^U = (L_k + 3U_k)/4$, and the current model $x^{(t)}$. Client $k$ trains two local models (one per rate) and returns the deltas. By linearly combining these deltas, the server aggregates the models corresponding to **all** $2^K$ lower/upper choices, thereby evaluating the rewards of the $2^K$ child nodes in one round.

The optimistic score of $\nu$ is

$$B(\nu) = V(\nu) + \tau \, \mathrm{diam}(\nu)^h + \sqrt{\frac{\alpha \ln(t+1)}{N(\nu)}}, \tag{23}$$

where $\mathrm{diam}(\nu)$ is the diameter of $\mathcal{I}(\nu)$ and $\tau, \alpha > 0$ are hyperparameters. The second term of $B(\nu)$ is borrowed from UCB bandits (Auer et al., 2002) exploration. In FedHOO, server also keep track of reward $V(\nu)$. Finally, the server expands $\nu$ into its $2^K$ child nodes corresponding to all combinations of two rates per client, and sets $x^{(t+1)}$ to the one with the highest reward among the $2^K$ candidates just evaluated.

Doubling each client's local training time unlocks an exponential exploratory gain. Enumerating those combinations explicitly would be prohibitive, but FedHOO obtains the same information at linear cost by utilizing parallelism of FL, making it vastly more efficient than existing search methods.

The strategy is especially advantageous in small federations, a configuration frequently encountered in industrial deployments. We therefore apply FedHOO when the number of clients is low. When $K$ is large, we revert to a uniform global sampling-rate because significant exploration cannot be completed within a reasonable training budget and a single rate promotes update alignment, a consideration that becomes increasingly critical as $K$ grows. We present the pseudocode of FedHOO in Algorithm 3. We write $\mathrm{diam}(\nu) = \max_k \left( U_k(\nu) - L_k(\nu) \right)$ corresponding to the $\ell_\infty$ width of the interval and use $\odot$ as element-wise product.

## B.2. Intuition with an Example

To illustrate how FedHOO works, consider a simple setting with $K = 2$ clients, and the search space is $[0, 1]^2$. The root node of the search tree corresponds to the full box $[0, 1] \times [0, 1]$. If we split this root into four quadrants, the child nodes are

$$[0, 0.5] \times [0, 0.5], \quad [0, 0.5] \times [0.5, 1.0], \quad [0.5, 1.0] \times [0, 0.5], \quad [0.5, 1.0] \times [0.5, 1.0].$$

---

**Algorithm 3** FedHOO

---

**Require:** bounds $r_{\min}, r_{\max}$, global rounds $T$, optimism constant $\alpha$
1: **initialize** $\nu_{\text{root}}$ with interval $\mathcal{I}(\nu_{\text{root}}) = [r_{\min}, r_{\max}]^K$; set $V(\nu_{\text{root}}) = 0$, $N(\nu_{\text{root}}) = 0$, $B(\nu_{\text{root}}) = +\infty$
2: **initialize** global model $x^{(0)}$
3: **for** $t = 0, \dots, T - 1$ **do**
4: $\quad \nu \leftarrow \arg\max_{\text{leaf}} B(\nu)$; let $\mathcal{I}(\nu) = \prod_{k=1}^{K}[L_k, U_k]$
5: $\quad$ **broadcast** $x^{(t)}$ and send $r_k^L = (3L_k + U_k)/4$, $r_k^U = (L_k + 3U_k)/4$ to each client $k$
6: $\quad$ **for each** client $k$ **in parallel do**
7: $\quad\quad$ train with rate $r_k^L$ starting from $x^{(t)}$ to obtain local update $\Delta_k^L = x^{(t)} - \bar{x}^L$; $\bar{x}^L$ is the resulting solution of the local training with $r_k^L$
8: $\quad\quad$ train with rate $r_k^U$ starting from $x^{(t)}$ to obtain local update $\Delta_k^U = x^{(t)} - \bar{x}^U$; $\bar{x}^U$ is the resulting solution of the local training with $r_k^U$
9: $\quad\quad$ **return** $(\Delta_k^L, \Delta_k^U)$
10: $\quad$ **end for**
11: $\quad$ **for each** $s \in \{0, 1\}^K$ **do**
12: $\quad\quad$ Define $\hat{r}^{(s)} = (1 - s) \odot L + s \odot U$ and $\tilde{r}^{(s)} = s \odot L + (1 - s) \odot U$
13: $\quad\quad \Delta^{(s)} = \sum_{k=1}^{K} p_k \big[(1 - s_k)\Delta_k^L + s_k \Delta_k^U\big]$
14: $\quad\quad x^{(s)} \leftarrow x^{(t)} - \Delta^{(s)}, \quad R^{(s)} \leftarrow \text{Validate}(x^{(s)})$
15: $\quad\quad$ create leaf $\nu_s$ with parent node $\nu$ with $I(\nu) = \Pi_k[\hat{r}_k^{(s)}, \tilde{r}_k^{(s)}]$
16: $\quad\quad$ set $V(\nu_s) \leftarrow R^{(s)}$, $N(\nu_s) \leftarrow 1$, $B(\nu_s) \leftarrow V(\nu_s)$
17: $\quad$ **end for**
18: $\quad s^\star \leftarrow \arg\max_s R^{(s)}, \quad x^{(t+1)} \leftarrow x^{(s^\star)}$
19: $\quad \bar{R} \leftarrow 2^{-K} \sum_s R^{(s)}$
20: $\quad$ **for node** $\nu'$ on path from $\nu$ to root **do**
21: $\quad\quad N(\nu') \leftarrow N(\nu') + 2^K$
22: $\quad\quad V(\nu') \leftarrow V(\nu') + \dfrac{2^K}{N(\nu')}\big(\bar{R} - V(\nu')\big)$
23: $\quad\quad B(\nu') \leftarrow V(\nu') + \tau \, \text{diam}(\nu)^h + \sqrt{\alpha \ln(t + 1) \, / \, N(\nu')}$
24: $\quad$ **end for**
25: **end for**

---

Exploring further, the child $[0, 0.5] \times [0, 0.5]$ can itself be split into four sub-boxes such as $[0, 0.25] \times [0, 0.25]$ and $[0, 0.25] \times [0.25, 0.5]$, and so on. This recursive partitioning continues as the algorithm zooms in on promising regions.

In the standard HOO algorithm, only *one* node can be explored in each round. For example, at round 1, HOO would select just a single quadrant, say $[0, 0.5] \times [0.5, 1.0]$, and update its statistics based on a single evaluation. Over many rounds, this gradually builds information, but the search may proceed slowly because each evaluation provides feedback for only one region of the tree.

By contrast, FedHOO leverages the federated setting to explore many regions at once using parallelism of FL. Because each client performs two local runs (with two different probe rates), the server can synthesize outcomes corresponding to all corner combinations of the current interval. In the two-client example, this means that in a *single* round, FedHOO obtains rewards for all four quadrants simultaneously. For instance, at round 1, FedHOO evaluates the four children at depth $h = 1$:

$$[0, 0.5] \times [0, 0.5], \quad [0, 0.5] \times [0.5, 1.0], \quad [0.5, 1.0] \times [0, 0.5], \quad [0.5, 1.0] \times [0.5, 1.0].$$

This parallel evaluation dramatically accelerates the search. Rather than spending four rounds to cover these quadrants (as in HOO), FedHOO requires only one. As the depth increases, the same principle applies: four sub-intervals at depth $h = 2$ can be evaluated together by reusing the two local runs per client.

### B.3. Summary and Limitations

The main advantage of FedHOO is that the federated setting allows the server to combine a small number of local runs into exponentially many synthetic evaluations. In the two-client example, two runs per client yield four corner evaluations in each round, and more generally $2^K$ corners can be evaluated from only two runs per client. This exponential coverage greatly accelerates the search compared to classical HOO, which can only evaluate a single node per round.

FedHOO is not intended as a universal tuner for very large federations. Its main purpose is to exploit the small-client regime, which appears in industrial deployments such as our CDD benchmark. In this regime, the exponential candidate set is still manageable, and the ability to evaluate joint rate combinations is more valuable than independent local tuning. For large-scale FL, we recommend either using a uniform global rate or applying FedHOO at a coarser level, such as over client clusters rather than individual clients.

Another limitation of this approach is the overhead of validating $2^K$ candidate models at each round, which may become expensive for large $K$. In addition, extending the procedure to partial participation is not straightforward, since missing client updates can prevent consistent synthesis of all corners. These issues suggest that while FedHOO is powerful for moderate $K$ or cluster-level dimensions, further work is needed to make it scalable to very large federations.

## C. Detail of the Experiment Setting

In this appendix, we provide full details of our experimental setup, including datasets, model architectures, training hyperparameters, and federated protocols.

### C.1. Datasets and Models

We evaluate on five benchmarks. CIFAR-10-LT and CIFAR-100-LT are long-tailed variants of CIFAR-10/100 with imbalance rates 20 and 100, constructed via the protocol of Liu et al. (2019). Concretely, for imbalance rate $\xi$, the number of training samples in class $c \in \{0, \ldots, C-1\}$ (sorted by frequency) is

$$N_c = N_{\max}\, \xi^{-c/(C-1)},$$

where $N_{\max}$ is the majority-class size and $C$ is the number of classes. If a dataset has balanced classes, we randomly sample from each class $c$ to have $N_c$ samples to make an imbalanced dataset. Test sets remain unaltered to test the performance of algorithms to produce balanced performance.

Adult Income (Becker & Kohavi, 1996) is a binary classification task with original imbalance ratio 3.17 and additional settings of 10 and 20 obtained by subsampling the minority class. UNSW-NB15 (Moustafa & Slay, 2015) provides 175,341 network-flow records spanning ten classes with a natural imbalance of roughly 434. For Adult Income and UNSW datasets, we preserve the same class imbalance in the test splits as in their corresponding training sets to mirror real-world conditions. The proprietary Chip-Defect-Detection (CDD) corpus comprises of approximately 780k optical micrographs (224×224) from five semiconductor fabs, in which only 1.7% of samples contain defects. CDD dataset is split by factory into 70% train, 15% validation, and 15% test. To expose the class imbalance inherent in our dataset, Figure 4 reports the number of defect and non-defect samples recorded by each factory.

We train a ResNet18 (He et al., 2016) model with group normalization from scratch on all public image benchmarks. For tabular tasks we employ a four layer fully connected network with hidden dimensions of 32, 16, 8, and 2 and ReLU activations on the Adult Income data and we adopt the CNN–LSTM architecture of Pear & Kibria (2024), consisting of stacked one-dimensional convolutional filters feeding into an LSTM to capture temporal correlations in network-flow features for UNSW-NB15. For CDD, we use ResNet-34. Table 5 summarizes each architecture.

### C.2. Implementation and Training Details

All algorithms—FedAvg (SGD), FedProx, weighted cross-entropy, Ratio Loss, FedGraB, CLIMB, FedCGN, and Fed-CGNM—share the same training search strategy. We tune the initial learning rate over the set $\{0.4, 0.2, 0.1, 0.05, 0.01\}$ by grid search and then decay it with a cosine annealing curve that reaches $10^{-4}$ at the final round. Weight decay is selected from $\{10^{-4}, 10^{-3}\}$ and the batch size is selected from $\{32, 64, 128\}$. For FedCGNM, we set stabilizing constant $\delta = 0.1$ across all datasets since the algorithm is robust under the choice of $\delta$ values. We also test the group momentum coefficient $\beta$

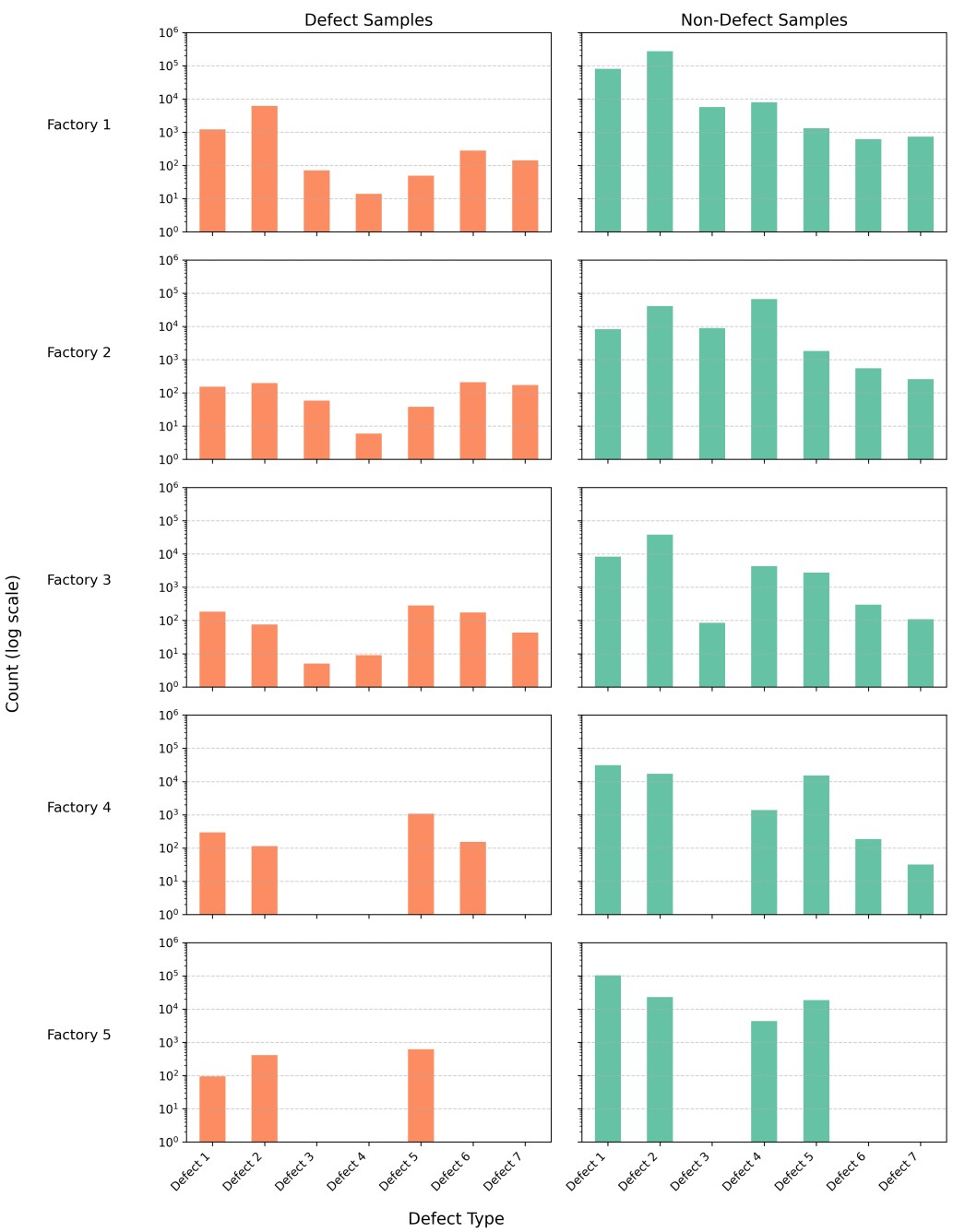

*Figure 4.* Distribution of sample counts by factory and defect code. Each row represents one factory; the left panel displays defect counts and the right panel shows non-defect counts, both on a logarithmic y-axis.

*Table 5.* Model specifications for each dataset used in our experiments.

| Dataset | Input | Backbone | Principal Layers |
|---|---|---|---|
| CDD | 224×224 RGB | ResNet-34 | conv $7 \times 7$–GN–ReLU; 4× residual stages |
| CIFAR-10-LT | 32×32 RGB | ResNet-18 | conv $3 \times 3$; 8× basic blocks |
| CIFAR-100-LT | 32×32 RGB | ResNet-18 | conv $3 \times 3$; 8× basic blocks |
| Adult Income | 104-dim tabular | 4-layer FFNN | $32 \to 16 \to 8 \to 2$ with ReLU and GN |
| UNSW-NB15 | 196-length sequence | CNN–LSTM | Conv1D[128,256,512] $\to$ LSTM $\to$ FC |

over $\{0.5, 0.6, 0.7, 0.8, 0.9\}$. Each communication round runs three local epochs when five clients participate and five local epochs when twenty clients participate on the image benchmarks; Adult Income and UNSW NB15 use a single local epoch per round because their training sets are comparatively small. All jobs are executed with PyTorch 2.4.1 and CUDA 11.4 on four NVIDIA TITAN Xp GPUs.

sampling-rates $r_k \in [0.4, 0.8]$ are tuned via FedHOO when $K = 5$, initializing a hierarchical tree over $[0.4, 0.8]^K$ with optimism constant $\alpha = 1.0$. In practice, since validation accuracy increases over rounds, we maintain each node's reward as an exponential moving average of validation accuracies for fair comparison. In each round, clients evaluate two candidate rates $(r_k^L, r_k^U)$, return update deltas, and the server extrapolates rewards for all $2^K$ combinations at linear cost. We run FedHOO only in the initial training phase and terminate it once the selected sampling-rates remain within the same per-client search intervals for five consecutive rounds. We then fix the resulting rates and use them for the remaining rounds to avoid long-term oscillation and validation overhead. For $K = 20$, a uniform global rate is used. In implementation, we perform three rounds of training without expanding tree to tackle cold start problem, since the initial training rounds are sensitive to sampling-rate choices, and cap the FedHOO search tree at depth 5. This provides a fine enough resolution of sampling-rates while keeping the exponential branching factor computationally manageable. Empirically, we found that deeper trees do not yield meaningful accuracy gains, as the noise inherent in federated training outweighs the benefit of additional granularity, whereas depth 5 provides a stable balance between exploration and efficiency.

In the small-scale regime ($K = 5$), all clients participate each round. In the moderate-scale regime ($K = 20$), we sample 50 % of clients per round. Public datasets are split IID/Non-IID across clients; the CDD corpus is partitioned by factory based on real meta data to simulate non-IID conditions.

## D. Further Discussion on Grouping Mechanism

This appendix provides additional discussion and ablation results that clarify the role of our grouping mechanism in handling class imbalance. The central idea is to balance two competing effects in multi-class optimization under skewed data. Treating each class independently can be overly sensitive to noisy, small-sample gradients and may introduce unstable update scaling, whereas grouping multiple classes can reduce variance by aggregating signals but may increase bias when heterogeneous classes are merged. The following subsections examine these trade-offs from complementary angles. We first show the instability of per-class normalization in practice. We then validate our variance-based threshold selection and study how the number of groups affects the bias–variance and overfitting behavior.

### D.1. Limitations of Per-Class Normalization (PCN)

Per-Class Normalization (PCN) rescales each class-specific gradient to unit norm. While this equalizes per-class magnitudes, it can induce unstable optimization dynamics and degrade generalization in multi-class regimes. First, aggregating $C$ unit-norm class gradients yields an update whose norm can vary widely from near 0 to as large as $C$, depending on the alignment among class directions. This introduces substantial scaling variability across iterations, which can impede stable convergence. Second, normalization does not reduce directional noise. When class-wise gradients fluctuate or are estimated from limited samples, PCN can amplify noisy directions and may overfit minority classes. Third, when $C$ is large, many classes are often absent from a mini-batch, and maintaining class-wise gradients becomes computationally and memory intensive.

Figure 5 provides empirical evidence of these issues in a centralized learning setting on CIFAR-10 with imbalance rate 20, using the same backbone (ResNet18) for both methods. Compared with SGD, PCN exhibits pronounced training instability with large loss spikes, while validation accuracy fluctuates and intermittently collapses, consistent with high directional noise and unstable effective step scaling.

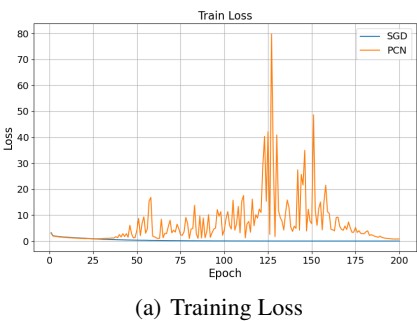

(a) Training Loss

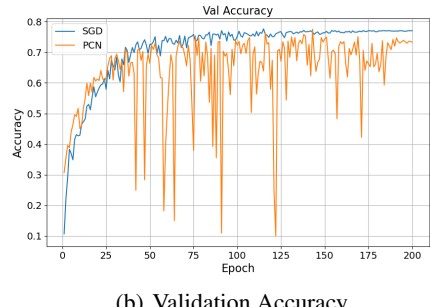

(b) Validation Accuracy

*Figure 5.* Instability of PCN in a multi-class setting in centralized learning setting. PCN shows large loss spikes and intermittent drops in validation accuracy, supporting that per-class normalization can yield unstable optimization and poor generalization when class-wise directions are noisy or poorly estimated.

## D.2. Validation of the Variance-Based Grouping Threshold

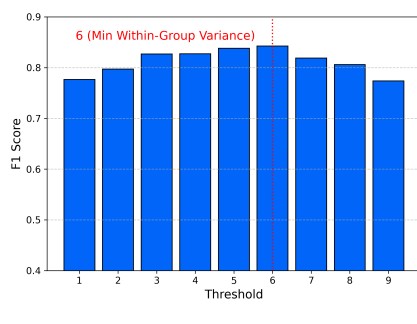

(a) CIFAR-10 - Imbalance rate=20

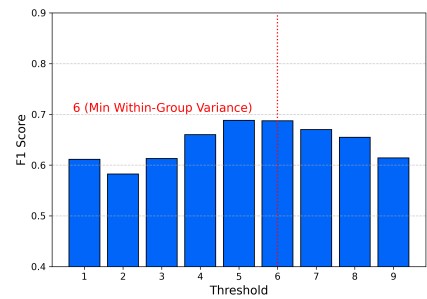

(b) CIFAR-10 - Imbalance rate=100

*Figure 6.* Test accuracy versus the number of rarest classes assigned to the minority group on CIFAR-10-LT. The dashed red line at $t = 6$ marks the threshold chosen by our variance-based grouping rule, which aligns with the peak test accuracy in both imbalance scenarios.

To confirm that our grouping rule reliably identifies the optimal split, we sweep the threshold $t \in \{1, \ldots, 9\}$, i.e. the number of rarest classes assigned to the minority group, and record test accuracy on CIFAR-10-LT under imbalance rates 20 and 100. As shown in Figure 6, the threshold $t = 6$ selected by minimizing within-group variance (vertical dashed red line) coincides with the highest performance in both settings, demonstrating that our data-driven rule matches the empirical optimum without exhaustive search.

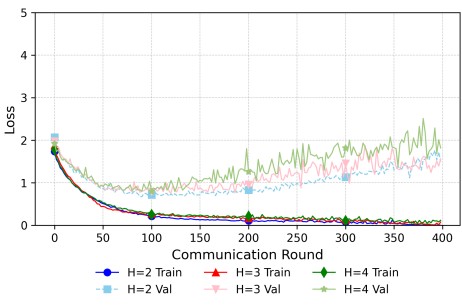

(a) CIFAR-10-LT, imbalance 20, $K = 5$

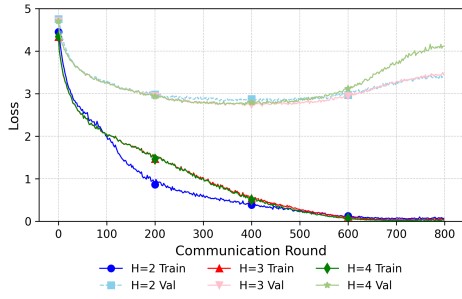

(b) CIFAR-100-LT, imbalance 20, $K = 20$

*Figure 7.* Training and validation loss for different numbers of groups $H \in \{2, 3, 4\}$.

## D.3. The Effect of the Number of Groups

To confirm that two groups offer the best bias–variance trade-off beyond the setting in Figure 3, we repeat the analysis on other settings. Figure 7 shows the training and validation losses when the number of groups $H$ is set to $\{2, 3, 4\}$ for CIFAR-10 and CIFAR-100.

In both benchmarks the pattern mirrors our earlier finding: validation loss stays lowest and most stable for $H = 2$, whereas $H = 3, 4$ starts to drift upward sooner, which is a signal of accelerated over-fitting. The training loss, by contrast, continues to fall for all values of $H$, which widens the train–validation gap when more than two groups are used. These results reinforce the conclusion that splitting classes into exactly two groups strikes the right balance between reducing gradient variance and controlling over-fitting for CIFAR-10 and CIFAR-100.

# E. Additional Experiment Results

In this appendix, we provide supplementary analyses to validate our design choices and assess the proposed algorithms under more challenging conditions.

## E.1. Effectiveness of FedHOO

Table 6 compares the company's metric obtained on the proprietary CDD benchmark when each federated optimizer runs with a fixed uniform resampling-rate versus when per-client rates are tuned by our FedHOO strategy. We include this result to demonstrate that FedHOO remains effective even when integrated with alternative training paradigms. FedHOO consistently boosts performance, confirming that our method is beneficial even when combined with optimizers other than FedCGNM.

*Table 6.* Performance improvement on CDD benchmark with a uniform global sampling-rate versus the FedHOO.

| Algorithm | Global Rate | FedHOO | improvement % |
|---|---|---|---|
| FedAvg | 72.83 | 85.37 | 17.22 |
| Weighted CE | 76.58 | 87.20 | 13.86 |
| Ratio Loss | 77.68 | 87.59 | 12.76 |
| FedCGN | 85.37 | 88.25 | 3.37 |
| FedCGNM | 86.88 | 90.87 | 4.59 |

To further isolate the benefit of FedHOO as a sampling-rate tuner, we compare it against two local rate-selection alternatives in the small-client regime. The first is a FAST-type (Wang et al., 2023) local exploration strategy, where each client independently explores its own rate using a bandit-style reward based on local loss decrease. The second is a locally optimal strategy, where each client independently selects its best local rate according to local validation and the resulting client-wise rates are combined for FL training. These baselines capture natural alternatives that tune resampling-rates locally rather than jointly.

Tables 7 and 8 show that FedHOO consistently outperforms both alternatives across FL optimizers. This supports the central design motivation of FedHOO: in FL, the best client-specific rates cannot generally be recovered by optimizing each client independently, because the final global model depends on interactions induced by federated aggregation. FedHOO explicitly searches this joint combinatorial space, while local strategies ignore cross-client interactions.

*Table 7.* FedHOO versus local rate-selection alternatives on CIFAR-10 with LT20 under Non-IID partitioning ($\psi = 0.5$).

| Method | w/o FedHOO | FedHOO | FAST-type | Locally Optimal |
|---|---|---|---|---|
| FedAvg | 0.7845 | **0.8026** | 0.7363 | 0.7690 |
| FedProx | 0.7826 | **0.7966** | 0.7445 | 0.7664 |
| FedGraB | 0.8166 | **0.8213** | 0.7552 | 0.8001 |
| Weighted CE | 0.7622 | **0.7758** | 0.6892 | 0.7595 |
| Ratio Loss | 0.7904 | **0.8089** | 0.7057 | 0.7884 |
| FedCGNM | 0.8316 | **0.8381** | 0.7788 | 0.8152 |

*Table 8.* FedHOO versus local rate-selection alternatives on CIFAR-100 with LT20 under Non-IID partitioning ($\psi = 0.5$).

| Method | w/o FedHOO | FedHOO | FAST-type | Locally Optimal |
|---|---|---|---|---|
| FedAvg | 0.4150 | **0.4278** | 0.4056 | 0.4120 |
| FedProx | 0.4089 | **0.4285** | 0.3957 | 0.4054 |
| FedGraB | 0.3733 | **0.3852** | 0.3538 | 0.3648 |
| Weighted CE | 0.3480 | **0.3789** | 0.3229 | 0.3253 |
| Ratio Loss | 0.4093 | **0.4216** | 0.3658 | 0.3798 |
| FedCGNM | 0.4351 | **0.4427** | 0.4068 | 0.4026 |

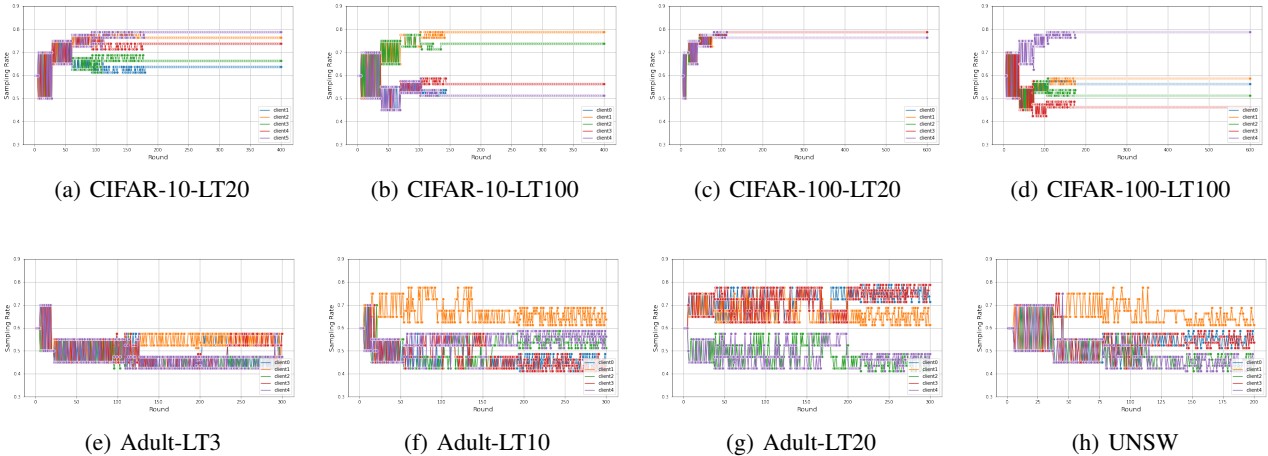

| | | | |
|---|---|---|---|
| (a) CIFAR-10-LT20 | (b) CIFAR-10-LT100 | (c) CIFAR-100-LT20 | (d) CIFAR-100-LT100 |
| (e) Adult-LT3 | (f) Adult-LT10 | (g) Adult-LT20 | (h) UNSW |

*Figure 8.* Per-client resampling-rate trajectories selected by FedHOO across datasets and imbalance settings.

## E.2. Resampling-rates in FedHOO

To better understand and illustrate the benefit of FedHOO, we plot and compare the trajectories of sampling-rate selection under FedHOO and standard HOO when each search method is run throughout training. In FedHOO, the selected sampling-rates correspond to the candidates yielding the best-performing model in each round, which is then adopted as the subsequent training initialization. As outlined in the implementation details, we restrict the search tree to a maximum depth of five.

Figure 8 illustrates how FedHOO adaptively selects per-client resampling-rates across different datasets and imbalance severities, and Figure 9 shows how HOO selects per-client resampling-rates on CIFAR-10-LT20 dataset. Each curve corresponds to one client, with the $y$-axis showing its selected sampling-rate, and the $x$-axis showing the communication rounds.

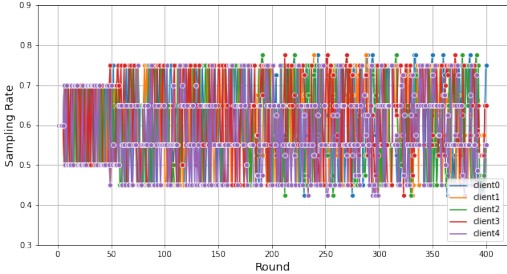

*Figure 9.* Per-client resampling-rate trajectories selected by standard HOO on CIFAR-10 iid setting with imbalance rate $\xi = 20$.

Under standard HOO, the algorithm explores only one branch of the search tree per round. As a result, the sampling-rate assigned to each client fluctuates heavily throughout training, and fail to converge in reasonable training time. The per-client trajectories remain noisy even after hundreds of rounds, reflecting the limited feedback that HOO gathers in each step.

In contrast, FedHOO leverages federated parallelism: by evaluating two candidate rates per client and linearly combining

their updates, it effectively observes all $2^K$ combinations at once. This parallel exploration dramatically accelerates the search. The sampling-rates quickly stabilize after the initial rounds, with each client settling into a distinct but consistent rate. The stabilized patterns observed across datasets in Figure 8 confirm that FedHOO both reduces variance and identifies effective configurations much earlier in training.

Overall, these plots highlight the key difference: while HOO suffers from slow and noisy adaptation due to sequential exploration, FedHOO achieves rapid and stable convergence by exploiting the structure of federated training.

### E.3. Alignment Effect of FedCGNM

Beyond mitigating directional noise, FedCGNM can also improve *client alignment*, which is a primary cause of performance degradation in federated learning (Dandi et al., 2022). Under heterogeneous and imbalanced data, clients often produce updates whose class-wise magnitudes differ substantially, which can amplify misalignment after aggregation. FedCGNM addresses this issue by normalizing each group update to a common unit scale, making client contributions more comparable and improving the alignment of aggregated directions.

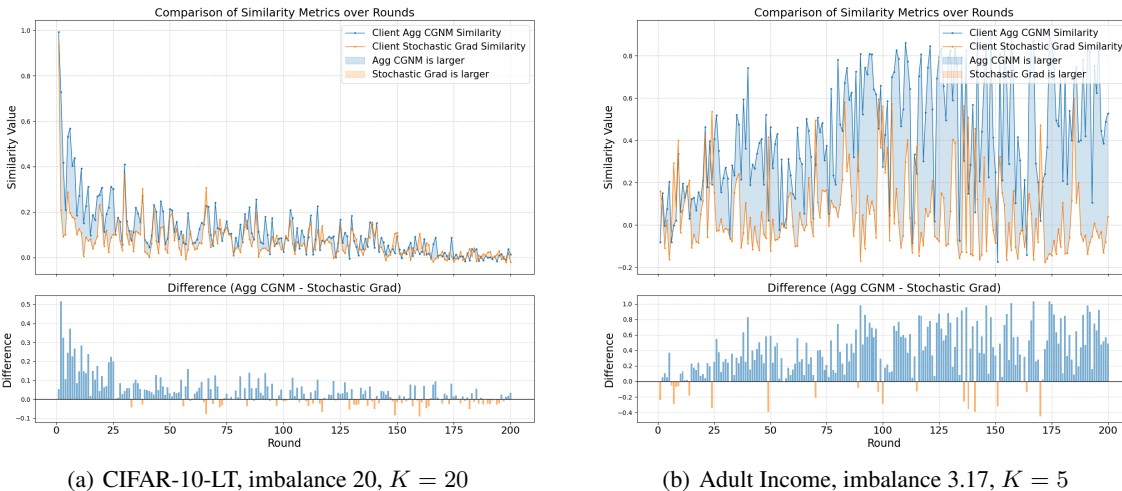

(a) CIFAR-10-LT, imbalance 20, $K = 20$        (b) Adult Income, imbalance 3.17, $K = 5$

*Figure 10.* Alignment comparison over rounds. The top panel plots similarity values for FedCGNM-induced updates versus stochastic-gradient updates, and the bottom panel shows their difference (FedCGNM minus stochastic gradient). Positive differences dominate, indicating improved alignment under FedCGNM.

To empirically assess this effect, we track an alignment metric over communication rounds and compare it against a stochastic-gradient baseline. At each round $t$, we compute the cosine similarity between a client update direction and a server's update direction (the corresponding aggregated update). We report (i) the similarity induced by FedCGNM updates and (ii) the similarity induced by stochastic-gradient updates under the same training pipeline. Figure 10 shows that FedCGNM typically achieves higher similarity across rounds, and the per-round difference (bottom plot) is predominantly positive, supporting that FedCGNM improves update alignment across clients.

### E.4. FedCGNM in Large-Federation Regimes

To further assess robustness, we evaluate FedCGNM under the challenging condition beyond the main experiments: large numbers of clients with partial participation.

**Large-client regime with partial participation.** We evaluate FedCGNM when the number of clients is large ($K = 100$) and only small fraction participate (10%) per round, which is a scenario that amplifies variance and typically harms optimization. We also use $E = 1$ for this scenario. Table 9 reports results on CIFAR-10-LT and CIFAR-100-LT with imbalance rates 20 and 100. While all methods degrade in this setting, FedCGNM consistently achieves the best performance, surpassing both reweighting-based and per-group gradient-normalization baselines.

Together, these results demonstrate that FedCGNM retains its advantage in more challenging federated environments: it scales to non-IID data and large-client regimes where baseline methods suffer the most.

*Table 9.* Performance in large-client federations with partial participation.

| Method | C10-LT20 | C10-LT100 | C100-LT20 | C100-LT100 |
|---|---|---|---|---|
| FedAvg + U | 0.3903 | 0.2710 | 0.0830 | 0.0471 |
| FedProx + U | 0.3901 | 0.2727 | 0.0874 | 0.0485 |
| Weighted CE + U | 0.4423 | 0.3131 | 0.0962 | 0.0533 |
| Ratio Loss + U | 0.3306 | 0.2266 | 0.0847 | 0.0464 |
| FedGraB + U | 0.4121 | 0.2971 | 0.0963 | 0.0345 |
| CLIMB + U | 0.4376 | 0.3305 | 0.1207 | 0.0655 |
| FedCGN + U | 0.4441 | 0.2994 | 0.1226 | 0.0696 |
| FedCGNM + U | **0.4675** | **0.3331** | **0.1227** | **0.0725** |

### E.5. Robustness under Various Data Heterogeneity

To further evaluate robustness under different degrees of client heterogeneity, we conduct additional experiments on CIFAR-100-LT20 with $K = 5$ clients. Table 10 reports the results under two heterogeneous data partitions. FedCGNM consistently outperforms the baselines, and FedHOO further improves performance in both settings. These results suggest that the proposed class-grouped normalized momentum remains effective under different degree of heterogeneity, while FedHOO provides additional gains by selecting client-specific resampling-rates.

*Table 10.* Final test accuracy on CIFAR-100-LT20 with $K = 5$ clients under different heterogeneity settings.

| Method | $\psi = 1.0$ | $\psi = 0.1$ |
|---|---|---|
| FedAvg + U | 0.3865 | 0.3654 |
| FedProx + U | 0.3921 | 0.3683 |
| Weighted CE + U | 0.3568 | 0.3444 |
| Ratio Loss + U | 0.3898 | 0.3625 |
| CLIMB + U | 0.4010 | 0.3804 |
| FedGraB + U | 0.3674 | 0.3169 |
| FedCGN + U | 0.4176 | 0.4121 |
| FedCGNM + U | 0.4252 | 0.4165 |
| FedCGNM + FedHOO | **0.4427** | **0.4423** |

### E.6. Per-Class Accuracy Distribution

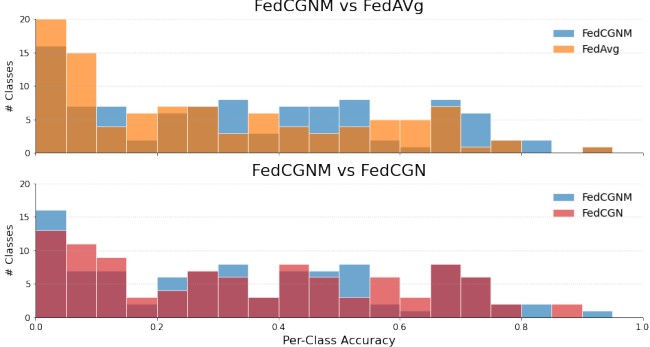

*Figure 11.* Distribution of per-class test accuracies on CIFAR-100-LT (imbalance rate=100, K=5).

From Figure 11, FedCGNM (blue) shifts the per-class accuracy distribution markedly to the right compared to FedAvg (orange), substantially reducing the number of near-zero classes. Whereas FedAvg produces a heavy tail of classes below 0.1 accuracy, FedCGNM elevates most classes into a moderate accuracy range. FedCGN also achieves a more balanced distribution, confirming the effectiveness of the grouping strategy in balancing class-wise performance.

### E.7. Empirical Diagnostics for Assumption 3.5

Assumption 3.5 excludes the degenerate case where the group-wise momentum is both nearly perfectly aligned with the corresponding gradient and nearly identical in magnitude to the scaled gradient. We empirically examine this condition by logging the two quantities appearing in the assumption: the gradient–momentum cosine similarity and the magnitude gap.

Figure 12 shows the results on AdultIncome and CIFAR-100-LT. On AdultIncome, the cosine similarity is often high, but the magnitude gap remains consistently positive and non-negligible. Thus, the magnitude-gap branch of the assumption is supported. On CIFAR-100-LT, the cosine similarity remains far below perfect alignment throughout training, and the magnitude gap is also positive. Thus, the low-alignment branch is strongly supported. These complementary behaviors suggest that Assumption 3.5 is mild in practice, indicating that the gradient and momentum are not simultaneously perfectly aligned and equal in magnitude along the observed training trajectories.

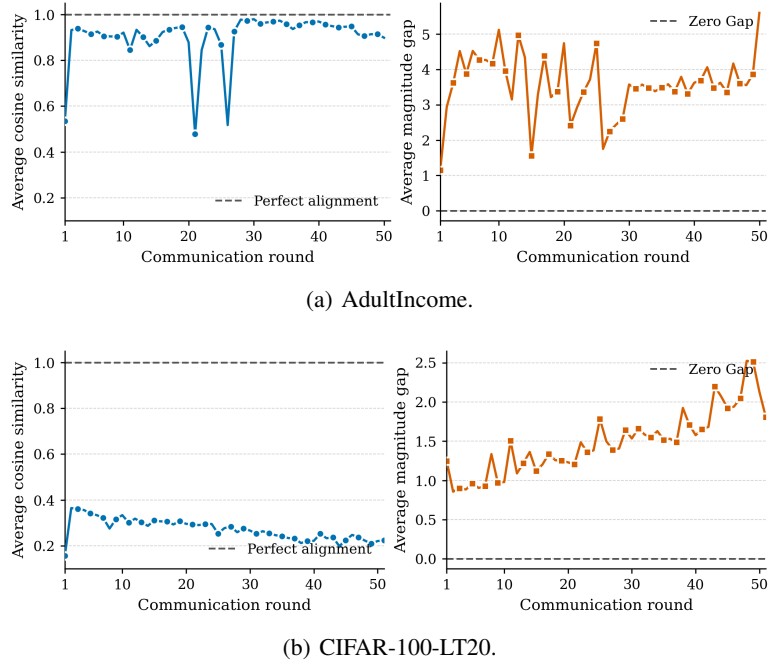

(a) AdultIncome.

(b) CIFAR-100-LT20.

*Figure 12.* Empirical diagnostics for Assumption 3.5. In both cases, the degenerate behavior does not appear in the observed trajectory.

