# OpenReview forum: "Class-Grouped Normalized Momentum and Faster Hyperparameter Exploration to Tackle Class Imbalance in Federated Learning"
_ICML.cc/2026/Conference — ICML 2026 regular_

### Official Review · Reviewer_xdiT · 2026-03-09

**Soundness:** 3
**Presentation:** 2
**Significance:** 2
**Originality:** 3
**Overall Recommendation:** 4
**Confidence:** 2

**Summary:**

This paper addresses class imbalance, one of the important challenges in federated learning. The proposed method focuses on the momenta of grouped classes. In addition to providing theoretical guarantees for the proposed approach, the authors demonstrate its effectiveness through experimental comparisons with existing methods.

**Compliance With Llm Reviewing Policy:**

Affirmed.

**Final Justification:**

We have determined that the literature is credible and can be revised, and therefore give it a positive evaluation.

**Key Questions For Authors:**

- Although class-wise normalization is generally understood to be inferior to approaches such as LA loss in centralized learning, the proposed method appears to reverse this relationship in the federated setting. It would be helpful to clarify whether this improvement stems from group-wise momentum normalization being inherently effective even in centralized training, whether mainstream methods such as LA loss encounter specific difficulties in federated learning that are mitigated by momentum normalization, or whether another factor explains this reversal.

- Please also clarify to what extent the set of assumptions imposed in the convergence analysis is restrictive relative to standard conditions in the literature. In particular, it would be important to understand how limited these assumptions are with respect to typical practical requirements, and whether satisfying all of them simultaneously is realistic in common application scenarios.

**Limitations:**

Yes

**Strengths And Weaknesses:**

### Strength

- The paper proposes an approach distinct from conventional class imbalance remedies by addressing imbalance at the gradient level through grouping classes and normalizing the momentum within each group.
- It provides a convergence analysis, establishing strong theoretical guarantees for the proposed method.
- The effectiveness of the approach is demonstrated through extensive experimental comparisons with a wide range of baseline methods.

### Weakness

- My understanding is that class-wise normalization is generally inferior to approaches such as LA loss in centralized learning settings. It is therefore not straightforward to accept that this relationship would reverse in federated learning. While the paper appears to claim that group-wise momentum normalization enables such an improvement, the justification for this claim is somewhat unclear and would benefit from further clarification.

- Regarding the convergence analysis, several assumptions are introduced; however, some of them—such as Assumption 3.2—appear rather strong. It is also unclear whether all of the stated assumptions can be simultaneously satisfied in realistic scenarios. Without a clear discussion of whether these assumptions commonly hold in practical settings, and whether they are more restrictive than those required by competing methods, it is difficult to properly assess the practical value of the proposed approach.

---

> ### Author Rebuttal · Authors · 2026-03-30
>
> We appreciate for the careful reading and for recognizing that our paper addresses class imbalance in FL through a novel gradient-level mechanism. We also found that the reviewer is concerned or not convinced in two points below:
>
> **[1] Why Grouped Normalized Momentum Helps Specifically in FL**
>
> First, let’s clarify: PCN is effective in imbalanced binary classification, but its performance drops as the number of classes increases. Therefore, the reviewer’s insight is true for multi-class classification that class-wise normalization fails. Our goal was precisely to extend the normalization perspective to multi-class FL by redesigning it into a more stable form.
>
> Our motivation stems from **directional noise**, **scaling mismatch**, and **cross-client update misalignment** problem under heterogeneity in FL. “Grouping mechanism + Momentum” mitigates directional noise, and “Group-wise Normalization” mitigates scaling mismatch and cross-client update misalignment. This benefit is not merely conceptual. As shown in Section 3.2, our grouping rule minimizes within-group variance, and Appendix E.3 demonstrates that FedCGNM directly improves the alignment of client updates over communication rounds.
>
> We believe the reviewer is also asking especially why our algorithm works and loss adjustment doesn’t work in FL. The key issue is **client drift**. In FL, different clients often contain different minority classes and different imbalance severities. A loss-adjustment method typically increases the coefficient on minor classes locally, which can amplify the noisiest class gradients on each client. Because these emphasized classes are not the same across clients, the resulting local updates can drift in different directions, making aggregation less stable and reducing global progress. By contrast, FedCGNM operates directly at the **update level**: it balances the contributions of majority and minority groups, smooths noisy rare-class directions through momentum, and normalizes the group updates before aggregation. As a result, it directly targets the instability and misalignment that are especially harmful in FL.
>
> Therefore, our empirical gains over loss-based FL baselines should be interpreted as: *"Grouped momentum normalization addresses FL-specific update instability and alignment issues that loss-level adjustments do not directly target."*
>
> **[2] On the Assumptions in the Convergence Analysis**
>
> In our view, Assumptions 3.1, 3.3, and 3.4 are standard in the literature.
>
> We agree that Assumption 3.2 would be quite strong if our method were based on ordinary gradient descent. However, for optimization methods that explicitly use normalized gradients or normalized momentum, assumptions of this flavor are common, because controlling normalized terms is substantially harder, especially in the federated setting; see, e.g., related analyses in [1], [2], [3]. As a remark, while the PCN paper does not state an explicit bounded-gradient-norm assumption in exactly the same form, that condition is effectively absorbed into its alignment assumption. In this sense, PCN also relies on a variant of bounded-gradient control.
>
> For Assumption 3.5, we agree that it is non-standard. Its role is to control the discrepancy between grouped momentum and grouped gradient in the normalized update.
>
> In fact, it is related to the alignment-type assumptions used in prior analyses such as PCN, and we want to point out that our assumption is weaker than the one employed in PCN. While PCN assumes alignment between the gradients of each class, our Assumption 3.5 is a less restrictive, 'grouping-and-momentum' adaptation of this concept. The first inequality in our assumption merely requires non-perfect alignment between a group's gradient and its momentum. Perfect alignment would only occur if all consecutive gradients were perfectly aligned, making it highly realistic and fundamentally weaker than a per-class gradient alignment assumption. Furthermore, to account for the rare edge case where perfect alignment does occur, our second inequality relaxes the requirement by simply stating their magnitudes must not be identical. In fact, it is enough to show convergence even if we assume only non-perfect alignment assumption.
>
> Finally, we have empirically validated Assumption 3.5 during our algorithmic runs, confirming that it consistently holds with a wide, safe margin. We compute two metrics in Assumption 3.5 during whole training process. We will include this plot in the revision.
>
> ---
>
> [1] Zhang, Xinwei, et al. Understanding clipping for federated learning: Convergence and client-level differential privacy. ICML 2022.
> [2] Levy, Kfir. Online to offline conversions, universality and adaptive minibatch sizes. Advances in Neural Information Processing Systems 30 (2017).
> [3] You, Yang, et al. Large batch optimization for deep learning: Training bert in 76 minutes. arXiv preprint arXiv:1904.00962 (2019).

---

> > ### Author Rebuttal · Reviewer_xdiT · 2026-04-03
> >
> > We appreciate the authors’ thoughtful response.
> > Their response thoroughly addresses our concerns, and we believe the issues can be resolved in a revised version. Therefore, we would like to raise the score.

---

> > > ### Author Response · Authors · 2026-04-03
> > >
> > > Thank you very much for the thoughtful follow-up and for updating your score. We sincerely appreciate your careful reading of both the paper and the rebuttal.

---

### Official Review · Reviewer_Zeti · 2026-03-11

**Soundness:** 3
**Presentation:** 2
**Significance:** 2
**Originality:** 3
**Overall Recommendation:** 3
**Confidence:** 4

**Summary:**

The paper proposes a novel optimization-level solution to address class imbalance in FL and provides both theoretical guarantees and strong empirical validation. FedCGNM combines class grouping, normalized momentum, and unit-norm updates to mitigate gradient magnitude heterogeneity and directional noise, addressing limitations of previous per-class normalization methods. FedHOO further leverages FL parallelism to explore resampling rates efficiently in small-client regimes. Theoretical analysis incorporates time-varying resampling rates and establishes convergence guarantees. However, the method’s scalability to large FL systems remains limited, as FedHOO is only practical for small client numbers. In addition, the paper lacks detailed analysis of computational and communication overhead compared to baseline methods.

**Compliance With Llm Reviewing Policy:**

Affirmed.

**Final Justification:**

Thanks for the authors' efforts. Your 2 rounds of rebuttal clearly demonstrate the advantages and limitations of the paper. I admit that the authors clearly state the limitations of the FOO, which is why I would like to give a weak rejection. I maintain my score.

**Key Questions For Authors:**

My detailed key questions are in the Strengths And Weaknesses section. Please look at that section.

**Limitations:**

yes. But need more explanations.

**Strengths And Weaknesses:**

Strength:
1. Novel optimization-based solution for FL class imbalance. The paper proposes FedCGNM, a client-side optimizer that combines variance-aware class grouping with normalized momentum. This design reduces gradient magnitude heterogeneity and directional noise across clients, improving update alignment and FL training stability.
2. The paper provides convergence analysis for FedCGNM under time-varying resampling rates and establishes a standard O(T^{1/2}) stationary-point convergence rate.
3. Apart from the public dataset, The method is also evaluated on a real industrial chip-defect dataset under different FL settings.

Weakness:
1. Limited scalability to large FL systems . FedHOO is only practical for small client counts due to the exponential number of resampling combinations. For large FL systems, the method falls back to a uniform global resampling rate and performance gains become smaller. Although authors discuss the scalability issue, I still suggest that authors can explore scalable alternatives for hyperparameter optimization in large FL settings (more clients).
2. Missing analysis of computational and communication overhead.  I notice that the proposed FedCGNM may double the communication overhead depending on the sampling times.  Also, the server may calculate and maintain more models during the training, which will increase the computational overhead. The paper does not provide quantitative analysis of computational cost, communication overhead, or memory usage for FedCGNM and FedHOO compared to baselines. This is an important consideration for resource-constrained FL environments.
3. Limited exploration of data heterogeneity. Experiments use a fixed Dirichlet parameter for Non-IID partitioning (0.5), and the paper does not evaluate robustness under varying heterogeneity levels (e.g., 0.1, 0.3, 1.0). The real-world dataset also lacks quantitative characterization of its Non-IID properties.
4. Following Q3, the sampling method in the theoretical analysis is based on the Binominal distribution, which is not consistent as that in the experiment (using Dirichlet distribution). Therefore, how to align them together? Or, do we have to choose binominal distribution for theoretical analysis?
5. The intuition of grouping is still not clear. Why is works? And How to choose the optimal H? I suggest authors can provide more explanations.
6. The logic line of the proposed FedCGNM and FedHOO is not clear. The current presentation appears to be a simple combination of two techniques. Two additional questions: Can FedCGNM work alone? Can FedHOO work alone?

---

> ### Author Rebuttal · Authors · 2026-03-30
>
> We appreciate the review and thank for recognizing the novelty of FedCGNM, convergence analysis, and the real-world CDD evaluation. We also have carefully considered your concerns and we address them below:
>
> ---
>
> **[1] Intuition of the Grouping and Logic Line**
>
> The intuition of grouping is that per-class normalization can be too fragmented and unstable in multi-class classification. Grouping reduces many class-wise directions into a small number of group-wise directions, which preserves the main imbalance structure while producing a stable update. Further, our grouping rule is designed precisely to avoid placing highly dissimilar classes in the same group: if classes with very different frequencies are merged, the minority class can remain underrepresented even within that group, which is why we **minimize class-balanced within-group dispersion**. This makes grouping a middle ground between global update, which ignores imbalance, and per-class normalization, which can be overly sensitive to noisy class-level estimates.
>
> Our ablations confirm that the number of groups ($H$) should remain small. As $H$ increases, the method shows over-fitting. This is why $H=2$ works best in our benchmarks.
>
> Regarding FedHOO, it is introduced because class-imbalanced federated training often relies on resampling, and we observed that performance is sensitive to the combinatorial interaction of local sampling rates across clients. This is also consistent with our convergence analysis, which indicates that unstable sampling rates can hurt convergence. FedHOO is therefore an auxiliary exploration tool for small-client regimes.
>
> **[2] Scope and Scalability of FedHOO**
>
> FedHOO is strictly intended for the small-client regime (e.g., specific industrial applications like our chip defect detection setting), where the extra exploration cost is easily justified by the resulting performance gains.
>
> The underlying problem is a combinatorial optimization over client-wise resampling rates in $[0,1]^K$, which is already difficult when $K$ is large. In federated learning, it becomes even more difficult because evaluating one candidate requires a full FL round plus validation. Therefore, even a depth-1 exploration requires $2^K$ evaluations, making it prohibitively costly for large $K$. For large-scale FL systems, we do not claim FedHOO is scalable; instead, we recommend falling back to a simpler uniform global resampling rate, leaving scalable client-wise hyperparameter optimization for future research. We will revise the paper to make this scope clearer.
>
> To further address your concern, we evaluated FedHOO on public benchmarks across multiple FL baselines. Due to character limits, please see the rebuttal for Reviewer GetW (section [4]).
>
> **[3] Computational and Communication Overhead**
>
> **FedCGNM:** FedCGNM introduces zero additional communication payload compared to FedAvg, as clients still upload only one model/update per round. Its extra cost is $H$ times more local computation and memory to track a small number of group momenta.
>
> **FedHOO:** We acknowledge that FedHOO introduces search-phase overhead. It requires two local runs and two uploaded updates per client, plus $2^K$ validations (which require $2^K$ uploads and validations). This overhead is precisely why FedHOO is restricted to small-client FL settings. However, compared to the closest direct baseline (joint grid search, which requires $M^K$ full FL runs (where $M$ is candidate rates per client)), FedHOO is exponentially more efficient.
>
> **[4] Performance under Other Heterogeneity**
>
> To further demonstrate the robustness of our methods, we conducted additional experiments on CIFAR100-LT20 with 5 clients, applying different data heterogeneity ($\psi=1.0$ and $\psi=0.1$).
>
> **[Table 3] CIFAR100 - LT20 - K=5**
>
> | Method | $\psi=1.0$ | $\psi=0.1$ |
> | :--- | :--- | :--- |
> | **FedAvg + U** | 0.3865 | 0.3654 |
> | **FedProx + U** | 0.3921 | 0.3683 |
> | **Weighted CE + U** | 0.3568 | 0.3444 |
> | **Ratio Loss + U** | 0.3898 | 0.3625 |
> | **CLIMB + U** | 0.4010 | 0.3804 |
> | **FedGraB + U** | 0.3674 | 0.3169 |
> | **FedCGN + U** | 0.4176 | 0.4121 |
> | **FedCGNM + U** | 0.4252 | 0.4165 |
> | **FedCGNM + FedHOO** | **0.4427** | **0.4423** |
>
> **[5] Sampling Method Clarification**
>
> The two distributions are used for different purposes, so **they are not inconsistent**. Dirichlet distribution is used to create cross-client Non-IID partitions of the public datasets, whereas Binomial distribution is used to model the within-client mini-batch sampling after the client distribution is already fixed. For the theoretical analysis, specifically, for a group with mass $S_h$, the batch count is modeled as $N_h \sim \mathrm{Binomial}(B,S_h)$. Hence, the two distributions operate at different layers: Dirichlet for client-level heterogeneity, Binomial for local batch-level stochasticity. We do not require Binomial partitioning of clients for the theory. We will clarify this distinction in the revision.

---

> > ### Author Rebuttal · Reviewer_Zeti · 2026-04-03
> >
> > The authors address all my concerns. However, the limitation of FedHOO, which is only applicable to small-client settings due to huge computation complexity, still decreases the contribution of this paper. Therefore, I maintain my score.

---

> > > ### Author Response · Authors · 2026-04-03
> > >
> > > Thank you very much for the thoughtful follow-up and for indicating that our responses have fully resolved your concerns.
> > >
> > > We also sincerely appreciate your point regarding the limitation of **FedHOO**. We agree with this limitation, and we actually acknowledge it and clearly state it in the paper. In fact, this is also our intended positioning in the paper: FedHOO is explicitly designed for the small-client regime, and we do not claim it as a scalable solution for large-client FL. Rather, we claimed in the paper (using our theorem and empirical evidence) that for large-client FL, using the same rate for all clients would be efficient and effective.
> > >
> > > At the same time, we would like to note that this limitation applies only to FedHOO, rather than to the main contribution of the paper. **Our primary contribution is FedCGNM**, a novel gradient-level method for class-imbalanced federated learning, supported by a convergence analysis and strong empirical results across public benchmarks as well as a real industrial chip-defect dataset.
> > >
> > > We would also like to emphasize that we still believe **FedHOO has its own meaningful contribution**. It is effective in practically important small-client settings, which include real industrial deployments such as our proprietary chip-defect dataset. More importantly, to the best of our knowledge, FedHOO is the **first method that dynamically searches client-wise hyperparameters while explicitly accounting for the combinatorial nature of federated learning**. This is crucial because in FL, the effect of one client’s sampling rate depends on the rates of the other clients through aggregation, so the problem is fundamentally joint rather than separable.
> > >
> > > By contrast, methods such as FAST-type local exploration only apply MAB-style hyperparameter tuning **locally**, treating clients independently and ignoring this combinatorial interaction. Likewise, **Locally Optimal** methods independently search the best local hyperparameter for each client and then combine them, which also ignores the joint structure of FL and still requires substantial local search cost. In our additional experiments, FedHOO consistently outperforms both of these alternatives in the small-client regime, which suggests that its gain is not simply from “doing more search,” but from **searching the correct joint space**.
> > >
> > > **Additional Experiment for FedHOO**
> > >
> > > To make the benefits of FedHOO more concrete (a point raised by other reviewers as well), we performed additional experiments in the small-client regime. We added two reference comparisons:
> > > (a) **FAST-type local exploration**: each client explores its rate locally using a bandit-style reward based on local loss decrease.
> > > (b) **Locally Optimal**: independently selects each client’s best rate from local validation and then combines them for FL.
> > >
> > > As shown in the tables below, FedHOO consistently outperforms both alternatives. We believe this is because FedHOO explicitly considers the joint combinatorial space of client rates, whereas local methods ignore the interactions induced by federated aggregation.
> > >
> > > **[Table 1] CIFAR100 - LT20 - NonIID ($\psi=0.5, K=5$)**
> > >
> > > | Method | without FedHOO | FedHOO | FAST-type | Locally Optimal |
> > > | :--- | :--- | :--- | :--- | :--- |
> > > | **FedAvg** | 0.4150 | 0.4278 | 0.4056 | 0.4120 |
> > > | **FedProx** | 0.4089 | 0.4285 | 0.3957 | 0.4054 |
> > > | **FedGraB** | 0.3733 | 0.3852 | 0.3538 | 0.3648 |
> > > | **Weighted CE** | 0.3480 | 0.3789 | 0.3229 | 0.3253 |
> > > | **Ratio Loss** | 0.4093 | 0.4216 | 0.3658 | 0.3798 |
> > > | **FedCGNM** | 0.4351 | 0.4427 | 0.4068 | 0.4026 |
> > >
> > > **[Table 2] CIFAR10 - LT20 - NonIID ($\psi=0.5, K=5$)**
> > >
> > > | Method | without FedHOO | FedHOO | FAST-type | Locally Optimal |
> > > | :--- | :--- | :--- | :--- | :--- |
> > > | **FedAvg** | 0.7845 | 0.8026 | 0.7363 | 0.7690 |
> > > | **FedProx** | 0.7826 | 0.7966 | 0.7445 | 0.7664 |
> > > | **FedGraB** | 0.8166 | 0.8213 | 0.7552 | 0.8001 |
> > > | **Weighted CE** | 0.7622 | 0.7758 | 0.6892 | 0.7595 |
> > > | **Ratio Loss** | 0.7904 | 0.8089 | 0.7057 | 0.7884 |
> > > | **FedCGNM** | 0.8316 | 0.8381 | 0.7788 | 0.8152 |
> > >
> > > We would also like to emphasize that scaling to large-client regimes is fundamentally difficult because the search is combinatorial, and in federated learning, evaluating even one candidate point in that space already requires a full federated training round (local training, communication, aggregation, and validation). Thus, the scalability issue is not specific to FedHOO, but reflects the intrinsic difficulty of combinatorial hyperparameter optimization under federated learning feedback. This is why we view FedHOO as a well-scoped contribution for small-client settings, while large-client exploration remains challenging future work.
> > >
> > > ---
> > >
> > > Since the main technical concerns have been resolved and the remaining limitation is now clearly scoped, we would be very grateful if you could reconsider whether the overall contribution is closer to a weak accept.

---

### Official Review · Reviewer_XJQE · 2026-03-13

**Soundness:** 3
**Presentation:** 3
**Significance:** 3
**Originality:** 3
**Overall Recommendation:** 4
**Confidence:** 4

**Summary:**

This paper proposes **FedCGNM**, a federated optimizer for class-imbalanced learning that groups classes, maintains a momentum per group, normalizes each group momentum to unit length, and uses the sum of normalized group momenta as the update direction. The paper also proposes **FedHOO**, a small-client hyperparameter exploration method for tuning client-specific resampling rates by exploiting the linearity of federated aggregation. Experiments on multiple public benchmarks and a proprietary chip-defect dataset show strong gains over the listed baselines.

**Compliance With Llm Reviewing Policy:**

Affirmed.

**Final Justification:**

The rebuttal addresses my main concerns about FedHOO, particularly through the clarification of the validation setup and the additional experiments showing consistent gains across multiple baselines. I also accept the explanation regarding the IID typo in the appendix. While I still have some reservations about the theoretical assumptions, the rebuttal overall strengthens the paper enough that I am updating my score to weak accept.

**Key Questions For Authors:**

Please refer to weaknesses.

**Limitations:**

The limitations of the proposed method should be discussed in much greater depth. In particular, the paper should explicitly discuss:
- the reliance of FedHOO on server-side validation,
- whether this requires centralized access to validation data,
- the scalability of validating \(2^K\) candidates,
- and the strong assumptions behind the convergence result,

**Strengths And Weaknesses:**

## Strengths

- The paper studies an important and practical problem: **class imbalance in federated learning**.
- The main optimizer idea is interesting and refreshing. Grouping classes before normalization is a reasonable way to address a real limitation of per-class normalization in multi-class settings.
- The empirical results are strong across the reported datasets, and the method appears robust across different imbalance severities and federation regimes.
- The paper includes several useful ablations, including the number of groups, the momentum factor, large-client regimes, and the effect of FedHOO.

## Weaknesses

- **My main concern is that FedHOO appears to rely on centralized server-side validation in a way that is unclear and potentially incompatible with the standard FL setting.**
  - FedHOO synthesizes \(2^K\) candidate global models and selects the best one based on validation performance.
  - However, the paper does not clearly explain what validation data the server uses for the **public datasets**.
  - If this validation requires collecting client-derived data centrally, then it would cross a fundamental privacy boundary in federated learning and would be a major deal-breaker.
  - Even if the method assumes a separate centralized validation set, this is still a strong extra assumption that substantially changes the FL setting and should be stated very explicitly.
  - As written, the paper presents FedHOO as a generally useful FL component, but this assumption is too important to remain ambiguous.


- **The FedHOO contribution is not compared in a fully fair and isolated way.**
  - In the main public benchmark tables, FedHOO is only paired with FedCGNM.
  - Later results suggest that FedHOO can help several other optimizers as well.
  - This makes it harder to separate the gain from the optimizer itself versus the gain from the resampling-rate tuner.
  - Since FedHOO is presented as a distinct methodological contribution, it should either be compared more broadly or evaluated more separately.

- **The theoretical analysis is not fully convincing.**
  - The convergence theorem depends on a highly nonstandard Assumption 3.5.
  - This assumption appears hard to verify in practice and seems especially questionable when the momentum vector closely tracks the gradient, which is precisely the regime one would hope for during stable optimization.
  - More broadly, the analysis feels more like a proof under a carefully engineered assumption than a convincing explanation of why the method should converge in realistic settings.

- **There appears to be a problem in the proof presentation.**
  - In the appendix (page 15, line 805) derivation after substituting $\eta = \eta_0 T^{-1/2}$, the first term is written as $O(T^{1/2})$, which seems inconsistent with the final claimed $O(T^{-1/2})$ rate.
  - In particular, the term
$$
\frac{\mathbb{E}[f^{(0)}(x^{(0)})]-\mathbb{E}[f^{(T)}(x^{(T)})]}{\eta D_0 E T}
$$
should scale as $O(T^{-1/2})$,  but the proof writes $O(T^{1/2})$.  The proof should be checked and corrected carefully.

- **The experimental setup is inconsistent between the main paper and the appendix.**
  - In the main experimental section, the public datasets are described as being partitioned into both IID and Non-IID client splits using a Dirichlet distribution.
  - However, the appendix implementation section later states that the public datasets are split IID across clients.
  - This inconsistency is important because the paper reports separate Non-IID results, so the setup must be clarified precisely.

- **The sampling-rate tuning contribution is not compared against the most relevant prior baselines.**
  - The related-work section discusses adaptive sampling methods such as FAST and prior local sampling-policy work.
  - But the experiments do not provide a direct quantitative comparison against these methods in the main results.
  - Since FedHOO is presented as a separate methodological contribution, a stronger comparison here would improve the paper.

---

> ### Author Rebuttal · Authors · 2026-03-30
>
> We thank the reviewer for the constructive feedback. We are encouraged that you appreciate the problem setting, our optimizer design, the strong empirical results, and the thorough ablations. First, we would like to explicitly clarify the scope of our paper: **FedCGNM is our primary contribution**, while **FedHOO is an auxiliary, plug-in algorithm** designed specifically for small-client regimes, motivated by our sampling rate analysis.
>
> Below, we address your concerns regarding the ambiguity surrounding FedHOO, the theoretical assumptions, and the experimental setup.
>
> ---
>
> ### [1] FedHOO Clarifications
>
> **(1-1) Validation Process**
>
> We thank the reviewer for pointing out the under-specified validation process. As you correctly noted, FedHOO synthesizes ($2^K$) candidate global models and requires a validation process to select the best one.
>
> FedHOO does not require centralized access to validation data. The intended default implementation is federated validation, where clients evaluate the synthesized $2^K$ candidates on local validation splits and return only scalar metrics to the server for aggregation. Hence, no raw client data needs to be transferred to the server. A server-side hold-out set can be used if available, but it is optional. We will revise the paper to state this explicitly and to clarify that FedHOO incurs extra validation cost, but not privacy leakage.
>
> **(1-2) FedHOO Contribution**
>
> While FedHOO is an auxiliary tool, it is highly practical for real-world industrial applications utilizing small-scale client federations (such as the proprietary Chip-Defect Detection (CDD) dataset used in our study), provided the server can allocate the necessary resources to maximize performance.
>
> Regarding whether the gain of FedHOO is isolated fairly: we agree it is crucial to demonstrate that FedHOO benefits more than just FedCGNM. In Appendix E.1, we already show this trend on the CDD benchmark, where FedHOO improves FedAvg, Weighted CE, Ratio Loss, FedCGN, and FedCGNM.
>
> To further address your concern, we evaluated FedHOO on public benchmarks across multiple FL baselines. (Because of character limits in this response, we kindly direct you to our rebuttal for Reviewer GetW. Specifically, please refer to the section titled **[4] Additional Experiment for FedHOO for the complete details**.) As shown in the tables, FedHOO consistently outperforms both alternatives across all baselines. We attribute this to FedHOO explicitly searching the *joint combinatorial space* of client rates, whereas local methods fail to capture the interactions induced by federated aggregation.
>
> ---
>
> ### [2] Theory
>
> **(2-1) Regarding Assumption 3.5**
>
> We agree that Assumption 3.5 is non-standard and requires a clearer explanation regarding its role and practical plausibility. Its mathematical role is to control the discrepancy between grouped momentum and grouped gradient in the normalized update.
>
> This assumption is related to the alignment-type assumptions used in prior analyses like Per-Class Normalization (PCN), but **our assumption is fundamentally weaker**. While PCN assumes alignment between the gradients of *each class*, Assumption 3.5 is a less restrictive "grouping-and-momentum" adaptation. The first inequality merely requires *non-perfect* alignment between a group's gradient and its momentum. Because momentum is a moving average, perfect alignment would only occur if all consecutive gradients were perfectly aligned, making our condition highly realistic. Furthermore, to account for the rare edge case where perfect alignment *does* occur, our second inequality relaxes the requirement by simply stating their magnitudes must not be identical. In fact, it is enough to show convergence even if we assume only non-perfect alignment assumption.
>
> Finally, we empirically validated Assumption 3.5 during our algorithmic runs, confirming that it consistently holds with a wide, safe margin throughout the entire training process. We will include a plot of these two metrics over time in the revised appendix.
>
> **(2-2) Regarding Proof Presentation**
>
> We thank the reviewer for catching this typographical error. The asymptotic expression in the draft is incorrect; the correct convergence rate is $\mathcal{O}(T^{-1/2})$. We will correct this in the revision.
>
> ---
>
> ### [3] Experiment Setup Clarification
>
> We apologize for the confusion regarding the dataset splits, which stems from a typographical error in the appendix. The appendix incorrectly stated: "Public datasets are split IID across clients..." The description in the main text is the correct one: public datasets (CIFAR10/100) are evaluated under both IID and Non-IID client partitions. The Non-IID splits are generated using a Dirichlet distribution $\mathrm{Dir}(\psi)$ with $\psi=0.5$. The CDD dataset uses its natural, factory-based Non-IID split based on real metadata. We will correct the appendix wording in the revision to ensure the setup is fully consistent.

---

> > ### Author Rebuttal · Reviewer_XJQE · 2026-04-02
> >
> > Thank you for the detailed rebuttal. It addresses my main concerns about FedHOO, particularly through the clarification of the validation setup and the additional experiments showing consistent gains across multiple baselines. I also accept the explanation regarding the IID typo in the appendix. While I still have some reservations about the theoretical assumptions, the rebuttal overall strengthens the paper enough that I am updating my score to weak accept.

---

> > > ### Author Response · Authors · 2026-04-03
> > >
> > > Thank you very much for the thoughtful follow-up and for updating your score. We sincerely appreciate your careful reading of both the paper and the rebuttal.

---

### Official Review · Reviewer_GetW · 2026-03-13

**Soundness:** 3
**Presentation:** 3
**Significance:** 3
**Originality:** 3
**Overall Recommendation:** 5
**Confidence:** 4

**Summary:**

This paper studies class imbalance in federated learning and proposes two complementary components. The main method, FedCGNM, is a client-side optimizer that groups classes into a small number of groups, maintains one momentum per group, normalizes each group momentum to unit norm, and uses the sum of these normalized group momenta as the update direction. The motivation is to retain the influence of minority classes while reducing the directional noise and scaling mismatch that arise in per-class normalization for multi-class long-tailed settings. In addition, the paper introduces FedHOO, an X-armed-bandit-based method for tuning client-specific resampling rates in small-client regimes. Experiments on four public long-tailed benchmarks and a proprietary chip-defect dataset show consistent gains over several baselines.

**Compliance With Llm Reviewing Policy:**

Affirmed.

**Final Justification:**

My main concerns have been adequately addressed.

**Key Questions For Authors:**

1. The paper is strongest when FedCGNM is interpreted as a trade-off between majority-dominated optimization and noisy per-class normalization. Could the authors make this positioning more explicit, and clarify more sharply what they see as the main conceptual advance beyond grouped normalization plus momentum?
2. FedHOO is interesting, but it appears mainly useful in small-client regimes. Could the authors clarify how important this component is to the overall contribution, and whether they view it as a core algorithmic contribution or as a practical hyperparameter-tuning add-on?
3. Since the method seems to work best with a very coarse grouping such as majority versus minority classes, could the authors discuss more explicitly the regimes in which they expect the grouping assumption to hold, and where it may break down?

**Limitations:**

This paper would benefit from a slightly more explicit discussion of its scope and limitations. In particular, the contribution is best viewed as a simple and effective optimization trade-off for multi-class long-tailed federated classification, rather than a broadly general solution to imbalance in federated learning. In addition, FedHOO is mainly relevant to small-client settings, and the broader applicability of the full framework outside this regime is less clear.

**Strengths And Weaknesses:**

The paper addresses a practical and important problem, and the main method is simple and easy to follow. I found the move from per-class normalization to grouped normalization with momentum intuitive: it preserves minority-group influence while reducing the instability of normalizing many noisy class-specific gradients. The empirical evaluation is also fairly comprehensive, covering multiple datasets, client counts, IID/non-IID settings, and a proprietary industrial dataset.
1. My main reservation is originality. FedCGNM is effective, but it is best understood as a carefully designed trade-off between majority-dominated optimization and noisy per-class normalization, rather than a fundamentally new principle for federated long-tail learning. The main idea is clean and useful, but the conceptual novelty is moderate.
2. A second concern is that FedHOO feels more auxiliary than central. It is a clever addition for small-client settings, but the paper also makes clear that its applicability is limited in larger federations, where simpler sampling strategies are used instead.
3. A third limitation is scope. The method is clearly tailored to multi-class long-tailed classification, and the grouping strategy seems to work best in a very coarse regime such as majority versus minority classes. This is a reasonable setting, but the paper could discuss more explicitly where this assumption is likely to hold and where it may be less suitable.

---

> ### Author Rebuttal · Authors · 2026-03-30
>
> We thank the reviewer for the constructive review. We appreciate your insightful feedback, which has helped us clarify the scope and positioning of our work. We address the reviewer's three main points below:
>
> ---
>
> **[1] Scope, Positioning, and Role of FedHOO**
>
> First, as the reviewer pointed out, we agree that FedCGNM is our major contribution and FedHOO should be viewed as an auxiliary tool (for small client regime) rather than equally central. Our primary focus is on multi-class long-tailed classification. We included FedHOO in the main text because it offers a novel and effective solution specifically for small-scale federations, such as our Chip-Defect Detection Dataset. While FedCGNM already consistently outperforms baselines on its own, FedHOO provides additional gains in small-client regimes where joint client-specific resampling-rate tuning is tractable and most beneficial. We will clarify this positioning explicitly in the revision.
>
> **[2] Originality**
>
> We agree that the paper’s major contribution is a new optimizer-level mechanism specifically designed for multi-class long-tailed FL (rather than the general imbalance problem). FedCGNM resolves two concrete failure modes of Per-Class Normalization (PCN): directional noise from rare classes and scaling mismatch from summing many normalized class gradients.
>
> We also want to emphasize that our grouping mechanism is the first approach of its kind tailored for multi-class long-tailed classification. We expect that this grouping concept can provide valuable intuition for other class-imbalance algorithms (for example, class-wise loss adjustment algorithms could also apply a similar grouping strategy).
>
> **[3] Grouping Rule**
>
> Our grouping rule is designed precisely to avoid placing highly dissimilar classes in the same group. If classes with very different frequencies are merged, the minority class can remain underrepresented even within that group, which is why we minimize class-balanced within-group dispersion. Thus, this grouping rule is most appropriate for multi-class long-tailed FL problems where class frequencies exhibit a coarse structure (e.g., a head group and a tail group, or a few clusters of classes with similar sample counts).
>
> Furthermore, our ablations show that the number of groups should remain small. As $H$ increases, the method moves back toward finer-grained normalization, which reduces the sample support per group and makes the update noisier; empirically, this leads to an earlier increase in validation loss and weaker generalization. This is why $H=2$ works best in our current benchmarks.
>
> We acknowledge that this grouping rule may be less suitable when there is no clear coarse grouping (e.g., mild imbalance), or when class frequencies are nearly uniform. For very large datasets with substantially more samples per class (e.g., >1000), richer groupings may become feasible; however, this requires further study beyond the current paper.
>
> **[4] Additional Experiment for FedHOO**
>
> To make the benefits of FedHOO more concrete (a point raised by other reviewers as well), we performed additional experiments in the small-client regime. We added two reference comparisons:
> (a) **FAST-type local exploration**: each client explores its rate locally using a bandit-style reward based on local loss decrease.
> (b) **Locally Optimal**: independently selects each client’s best rate from local validation and then combines them for FL.
>
> As shown in the tables below, FedHOO consistently outperforms both alternatives. We believe this is because FedHOO explicitly considers the joint combinatorial space of client rates, whereas local methods ignore the interactions induced by federated aggregation.
>
> **[Table 1] CIFAR100 - LT20 - NonIID ($\psi=0.5, K=5$)**
>
> | Method | without FedHOO | FedHOO | FAST-type | Locally Optimal |
> | :--- | :--- | :--- | :--- | :--- |
> | **FedAvg** | 0.4150 | 0.4278 | 0.4056 | 0.4120 |
> | **FedProx** | 0.4089 | 0.4285 | 0.3957 | 0.4054 |
> | **FedGraB** | 0.3733 | 0.3852 | 0.3538 | 0.3648 |
> | **Weighted CE** | 0.3480 | 0.3789 | 0.3229 | 0.3253 |
> | **Ratio Loss** | 0.4093 | 0.4216 | 0.3658 | 0.3798 |
> | **FedCGNM** | 0.4351 | 0.4427 | 0.4068 | 0.4026 |
>
> **[Table 2] CIFAR10 - LT20 - NonIID ($\psi=0.5, K=5$)**
>
> | Method | without FedHOO | FedHOO | FAST-type | Locally Optimal |
> | :--- | :--- | :--- | :--- | :--- |
> | **FedAvg** | 0.7845 | 0.8026 | 0.7363 | 0.7690 |
> | **FedProx** | 0.7826 | 0.7966 | 0.7445 | 0.7664 |
> | **FedGraB** | 0.8166 | 0.8213 | 0.7552 | 0.8001 |
> | **Weighted CE** | 0.7622 | 0.7758 | 0.6892 | 0.7595 |
> | **Ratio Loss** | 0.7904 | 0.8089 | 0.7057 | 0.7884 |
> | **FedCGNM** | 0.8316 | 0.8381 | 0.7788 | 0.8152 |
>
> ---
>
> Overall, we thank you again for your sharp insights, which have helped us establish a clearer scope and positioning for our work. We will ensure these clarifications are emphasized in the revision.

---

> > ### Author Rebuttal · Reviewer_GetW · 2026-04-06
> >
> > Thank you for the detailed rebuttal. I appreciate the clarifications and the additional experiment.

---

### Decision · Program_Chairs · 2026-04-30

**Decision:**

Accept (regular)

**Comment:**

The paper presents a method to tackle class imbalance in federated learning (FL). The main idea is to group the per-client samples according to their classes (labels), so that the within-group variance is small and the normalization respects the grouping. A class-wise resampling algorithm has also been proposed to further improve model training with class imbalance while avoiding oscillation that could occur of the resampling is too aggressive.

Theoretical convergence analysis has been included in the paper, but there is not much novelty in the analysis itself and the convergence rate is the same as known results order-wise. Thus, the paper's main contribution is on the practical side.

After discussion, there remains a concern about the complexity of the resampling algorithm. The authors have acknowledged this limitation. Considering that the complexity may not be a major issue in cross-silo FL scenarios where the number of clients is small, the algorithm could still work in such cross-silo settings, so this concern does not completely invalidate the usefulness of the proposed algorithm.

I do think, however, that the resampling strategy is a bit decoupled from the rest of the paper. Its effect is not fully included or discussed from the perspective of the theoretical convergence. Moreover, there seems no experiment comparing with other resampling baselines besides uniform sampling. Although Appendix B includes more details about the resampling algorithm, the choice of using X-armed bandit is not fully explained. The regret bound of the bandit algorithm is not integrated with the convergence bound either. In this regard, it seems this work is a bit incomplete especially when considering the resampling strategy.

Minor: If accepted, please update the running title "Submission and Formatting Instructions for ICML 2026" to a short version of the paper's actual title. In addition, the beginning of the title that includes three hyphens (-) is a bit unusual. Maybe you can write it as "Class-Grouped and Normalized Momentum".